EMBO
reports

# G-Quadruplexes act as sequence-dependent protein chaperones

Adam Begeman[1,†] ⓘD, Ahyun Son[1,†] ⓘD, Theodore J Litberg[1] ⓘD, Tadeusz H Wroblewski[1], Thane Gehring[1], Veronica Huizar Cabral[1], Jennifer Bourne[2], Zhenyu Xuan[3] & Scott Horowitz[1,*] ⓘD

## Abstract

**Maintaining proteome health is important for cell survival. Nucleic acids possess the ability to prevent protein aggregation more efficiently than traditional chaperone proteins. In this study, we explore the sequence specificity of the chaperone activity of nucleic acids. Evaluating over 500 nucleic acid sequences' effects on protein aggregation, we show that the holdase chaperone effect of nucleic acids is sequence-dependent. G-Quadruplexes prevent protein aggregation via quadruplex:protein oligomerization. They also increase the folded protein level of a biosensor in *E. coli*. These observations contextualize recent reports of quadruplexes playing important roles in aggregation-related diseases, such as fragile X and amyotrophic lateral sclerosis (ALS), and provide evidence that nucleic acids have the ability to modulate the folding environment of *E. coli*.**

**Keywords** nucleic acids; protein aggregation; protein folding; proteostasis; RNA
**Subject Categories** RNA Biology; Translation & Protein Quality

See also: **J Aarum** *et al* (October 2020)

## Introduction

Chaperones are a diverse group of proteins and other molecules that regulate proteostasis (Hartl *et al*, 2011) in the cell by preventing protein aggregation (holdases) and helping protein folding (foldases). Recently, molecules other than traditional protein chaperones have been shown to play important roles in these processes (Gray *et al*, 2014; Ray, 2017). We recently showed that nucleic acids can possess potent holdase activity, with the best sequences having higher holdase activity than any previously characterized chaperone (Docter *et al*, 2016). Nucleic acids can also collaborate with Hsp70 to help protein folding, acting similarly to small heat

shock proteins (Jakob *et al*, 1993, 1999; Haslbeck & Vierling, 2015; Docter *et al*, 2016). Nucleic acids can also bring misfolded proteins to stress granules (Bounedjah *et al*, 2014) and are a primary component of the nucleolus, which was recently shown to store misfolded proteins under stress conditions (Frottin *et al*, 2019). However, the structural characteristics, sequence dependence, and mechanistic understanding of how nucleic acids act as chaperones remain unclear.

A critical question in understanding the holdase activity of nucleic acids is whether this activity is sequence specific? Previously, we showed that polyA, polyT, polyG, and polyC prevented aggregation with varying kinetics, suggesting that sequence specificity is possible (Docter *et al*, 2016). Here, we tested sequence specificity by examining over 500 nucleic acids of varying sequence for holdase activity. The holdase activity is found to be highly sequence specific, with quadruplexes showing the greatest activity. Several quadruplexes displayed generality, with potent holdase activity for a variety of different proteins. This activity was also verified to occur in *Escherichia coli*. Further examination of these quadruplex sequences demonstrated that the holdase activity largely arises through quadruplex:protein oligomerization. These results help explain several recent reports of quadruplex sequences playing important roles in oligomerization, aggregation, and phase separation in biology and pathology, and that these are common properties of quadruplex interactions with partially unfolded or disordered proteins. They also represent a strong demonstration of the ability of nucleic acids in modulating protein folding in *E. coli*.

## Results

### Sequence specificity of holdase activity

To determine the sequence specificity of the holdase activity of nucleic acids, we measured light scattering and turbidity via absorbance in a thermal aggregation assay (Fig 1A) for 312 nucleic acid sequences (Fig 1B). These nucleic acids were nearly all 20 bases in length, single-stranded DNA (ssDNA) sequences of random composition. Bulk DNA was used as a positive control (Docter *et al*, 2016).

1 Department of Chemistry & Biochemistry, Knoebel Institute for Healthy Aging, University of Denver, Denver, CO, USA
2 Department of Cell and Developmental Biology, University of Colorado School of Medicine, Aurora, CO, USA
3 Department of Biological Sciences, Center for Systems Biology, University of Texas at Dallas, Richardson, TX, USA
*Corresponding author. Tel: +1 303 871 4326; E-mail: scott.horowitz@du.edu
†These authors contributed equally to this work

Plotting the percent aggregation for each sequence demonstrates that the holdase activity of the ssDNA is sequence-dependent (Fig 1B). Sequences nearly spanned the complete range in activity, from barely affecting aggregation, to nearly preventing all protein aggregation for over an hour.

With this high level of sequence dependence, we next performed bioinformatics to determine if any sequence motifs encoded holdase activity. We first found that the holdase activity is positively correlated with the guanine content in the sequences ($\rho = 0.24$, $P$-value = $1.5 \times 10^{-5}$). Comparing the top third in holdase activity to the bottom third, only one motif was found to be significantly enriched in sequences with higher holdase activity (53.85% vs. 7.69%, FDR = 0.001; Fig 1C). This motif contains five consecutive guanines followed by any base and then thymine. A similar guanine-rich (G-rich) motif (consensus pattern: BGGSTGAT) was also found by a regression-based method ($R^2 = 0.61$, $P$-value = $1 \times 10^{-5}$). This analysis suggests that the most potent holdase activity was encoded by a polyG motif.

To verify this polyG motif, we tested another 192 sequences for holdase activity that had high guanine content. These sequences include 96 sequences with a 55% bias toward guanine bases, 40 sequences with a 75% bias toward guanine bases, and 56 having different positional variations of the aforementioned polyG motif (Fig 1C). Comparing the original random sequences to these G-rich sequences, the average aggregation was substantially reduced in the enriched guanine set, from 64.8% to 32.0%. Within the enriched guanine set, however, there was still a great deal of variation, with the data spanning aggregation from 72% to 4%. This wide variability suggests that the motif requires more than just the high guanine content. Within the subset of sequences with a 55% bias toward guanine, a significant polyG motif was again identified by comparing sequences having different holdase activity. Within the subset of 75% guanine-containing sequences, no statistically significant differences were found, as most sequences contained at least one polyG motif. We also tested holdase activity for 56 sequences having different positional variations of the aforementioned polyG motif, which did not find positional dependency within the sequence for the holdase activity. In the best sequences from this enriched assay, the nucleic acid completely prevented protein aggregation for the entire hour and a half experiment.

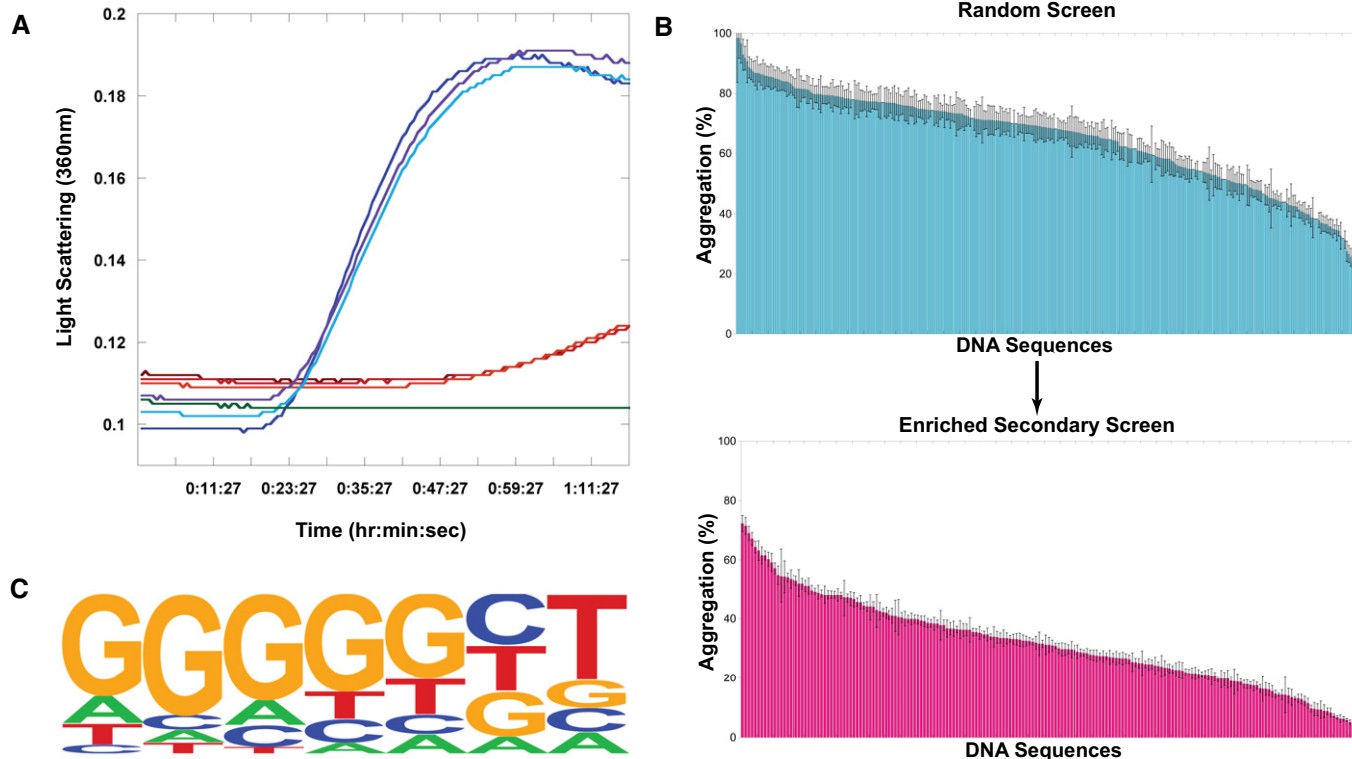

**Figure 1. Testing Sequence Dependence of Chaperone Nucleic Acid Activity.**

A  Representative example citrate synthase protein aggregation assay. Turbidity and light scattering were measured in a multimode plate reader at 360 nm for 1.5 h of incubation at 50°C. Blue lines represent triplicate citrate synthase alone, red and orange lines represent triplicate citrate synthase incubated with a single ssDNA sequence, green is buffer alone.
B  Screen of ssDNA sequences for holdase chaperone activity. Each bar represents a different 20-nt sequence, sorted by activity. Aggregation % was measured as the normalized average of triplicate (technical replicate) citrate synthase turbidity measurements after 1.5 h of incubation at 50°C (representative example shown in Fig 1A). Lower aggregation indicates greater holdase function. The initial screen used random, non-redundant sequences (top), which led to a follow-up enriched screen (bottom). Error bars are mean ± SE.
C  HOMER Logo of motif found by analyzing screen (statistics using a binomial distribution with the default setting by HOMER to calculate $P$-value of motif enrichment (Benner *et al*, 2017): $P < 1.0 \times 10^{-13}$, FDR < 0.001, % of Targets: 53.85, % of Background: 7.69).

Source data are available online for this figure.

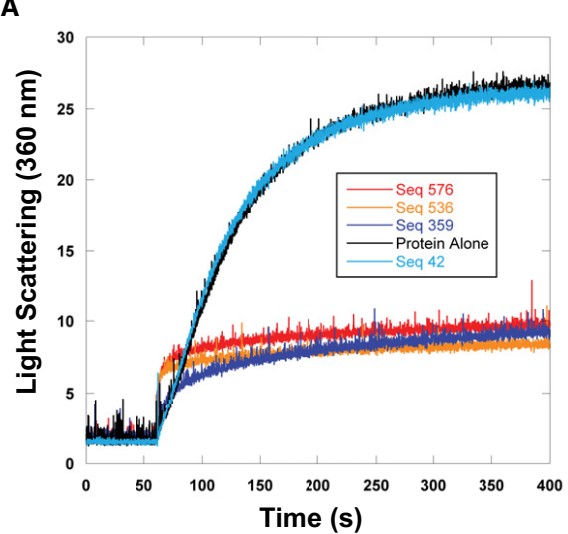

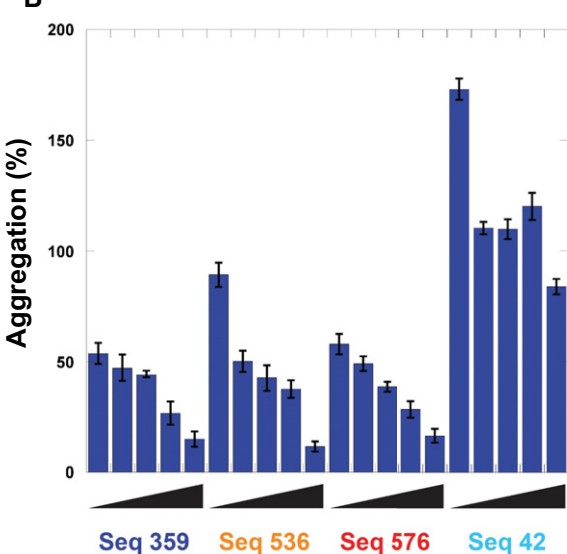

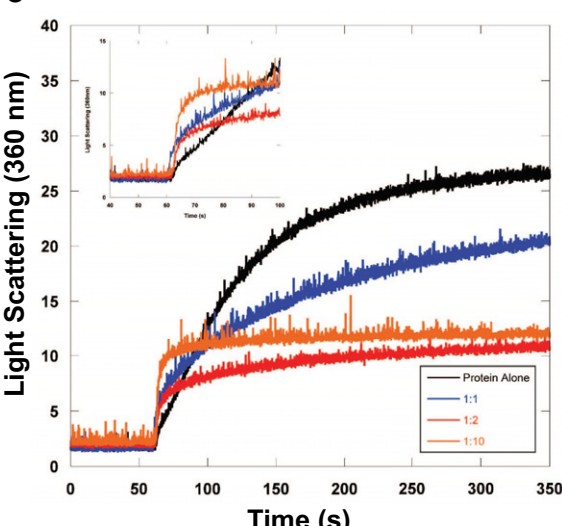

Figure 2. **Concentration dependence of chaperone nucleic acid activity.**

A  Citrate Synthase aggregation from chemically induced denaturation via right angle light scattering at 360 nm. Sequences 359, 536, and 576 all displayed holdase activity and contain a polyG motif. Sequence 42 was used as a negative control, as it performed poorly as a holdase chaperone and did not contain a polyG motif.

B  Percent aggregation in thermal denaturation assay with varying concentrations of select quadruplex-forming sequences (Sequences 359, 536, and 576) and negative control sequence (Seq42). Concentrations are ssDNA strand to protein ratios of: 0.5:1, 1:1, 2:1, 4:1, and 8:1. Error bars are mean $\pm$ SE of technical triplicates.

C  Citrate Synthase aggregation from chemically induced denaturation via right angle light scattering at 360 nm as a function of concentration of Sequence 359.

Source data are available online for this figure.

These results confirmed that the holdase activity is associated with a polyG motif.

We further characterized the holdase activity of the polyG-containing sequences using chemical denaturation aggregation assays in which the protein starts in a denatured state. Light scattering experiments confirmed the holdase activity in at least nine different polyG-containing sequences (Figs 2A and EV1, and Table 1). These data suggest that the polyG-containing sequences preferentially bind to a misfolded or partially denatured form of the protein rather than the native state. Choosing three of these sequences for further investigation (sequences 359, 536, and 576, Table 1), we performed concentration dependent assays and found that all three displayed strong concentration dependence in their activity (Fig 2B and C), unlike a control sequence with no polyG motif (sequence 42, Table 1), which did not display strong activity in our initial screen.

## G-Quadruplexes as potent holdases

PolyG is well known to form quadruplexes when provided with appropriate counter ions. Composed of polyG bases forming pi-stacked tetrads, guanine quadruplexes are a class of structured nucleic acids that have been of increasing interest due to their regulatory role in replication, transcription, and translation (Paeschke et al, 2005; Rhodes & Lipps, 2015). Quadruplexes have also recently been implicated in several protein aggregation genetic disorders, such as fragile X syndrome and ALS (DeJesus-Hernandez et al, 2011; Zhang et al, 2014; Stefanovic et al, 2015; Vasilyev et al, 2015; Afroz et al, 2017; Balendra & Isaacs, 2018).

To test if the sequences containing polyG that had potent holdase activity were forming quadruplexes, we performed circular dichroism (CD) spectroscopy experiments on sequences 359, 536, and 576 to determine their secondary structure. The CD spectra showed distinct peaks at 260 and 210 nm, with a trough at 245 nm, indicative of parallel quadruplex formation (Fig 3A; Paramasivan et al, 2007). This pattern was not observed in sequence 42, which had no polyG motif, and was therefore not expected to form a quadruplex. This supposition was further supported by examining the emission spectra of N-methylmesoporphyrin IX (NMM), a well-characterized parallel quadruplex-binding fluorophore (Sabharwal et al, 2014; Manna & Srivatsan, 2018). Due to the sensitivity limitations of CD, we examined the fluorescence spectra of NMM in the presence of

**Table 1.  Select DNA sequences used in this study.**

| Name | Sequence | Aggregation Assays in Figures | Quadruplex? (G4 Hunter Score or reference) |
|---|---|---|---|
| Seq359 | GGG GGG GTA ACG GGC TGG TT | Figs 2 and 4 (SeqA[a]), EV1, EV4 | Yes (1.95) |
| Seq536 | GAG GGG GGC TGC CGT TCA CA | Figs 2 and 4 (SeqF[a]), EV1 | Yes (1.00) |
| Seq576 | TGT CGG GCG GGG AGG GGG GG | Figs 2 and 4 (SeqJ[a]), Fig 4, EV1 | Yes (2.60) |
| Seq42 | AAC GAA AGA ACA TAA TCT CG | Figs 2, EV1 | No (< 0.1) |
| Seq398 | GGG GGG CGG TGC GGT AGC GA | Fig 4 (SeqB[a]) | Yes (1.60) |
| Seq567 | GGG GGG GCG GCG GGG GGG AG | Fig 4 (SeqC[a]) | Yes (2.95) |
| Seq361 | CGG ATG GGG TGG GTG CTG GA | Fig 4 (SeqD[a]) | Yes (1.60) |
| Seq573 | GGC GGG CGG TGG GGG GTG CG | Fig 4 (SeqE[a]), EV1 | Yes (2.00) |
| Seq563 | TAG GTG GGA GGT GCG GGA GG | Fig 4 (SeqG[a]) | Yes (1.50) |
| Seq357 | ATG AGT TGG TGC GTG GGG GA | Fig 4 (SeqH[a]) | Yes (1.35) |
| Seq345 | GGG GTT GGT GGG GGG GTA TA | Fig 4 (SeqI[a]) | Yes (2.40) |
| Seq582 | GGG GGG GAC GGT GGC GAG GG | Fig 4 (SeqK[a]) | Yes (2.20) |
| Seq592 | GGG GGG GGG GCC GGG GGG GT | Fig 4 (SeqL[a]), EV1 | Yes (3.20) |
| Seq579 | GGA GGG GGG GGG GTG AGG GG | Fig 4 (SeqM[a]) | Yes (3.05) |
| Seq353 | CGG CCG GGC GGG GTC CGG TT | Fig 4 (SeqN[a]) | Yes (1.15) |
| Seq364 | GGG CTT TGC ATT TCT ATG GT | Fig 4 (SeqO[a]) | No (0.55) |
| Seq63 | GTG GGA TGT CAG ACG TGG AC | Fig 4 (SeqP[a]) | No (0.7) |
| Seq60 | CAT CCG AGG TTT ACT CCC CC | Fig 4 (SeqQ[a]) | No (−1.05) |
| Seq185 | AAA CGT GCA GTG CAA CAT AA | Fig 4 (SeqR[a]) | No (< 0.1) |
| Seq190 | GTA CTT TTG GCA TCC TCA CA | Fig 4 (SeqS[a]) | No (−0.15) |
| Seq259 | AGT CTT GTT GTG ACT CAA CT | Fig 4 (SeqT[a]) | No (< 0.1) |
| Seq305 | GAT GAT GTC CGT AGC TTG CC | Fig 4 (SeqU[a]) | No (−0.15) |
| Seq347 | AAT GGG ATG CCA TTT GCT GG | Fig 4 (SeqV[a]) | No (0.5) |
| Seq209 | GAT ATA GCT GGA GTA CAA CC | Fig 4 (SeqW[a]) | No (< 0.1) |
| Seq205 | CCA CGA CTG CAG AGG TAT GT | Fig 4 (SeqX[a]) | No (< 0.1) |
| Seq340 | GGT AGT TCG GTT GGT GGG GA | Fig EV1 | Yes (1.40) |
| Seq589 | GCG GGG GGA GGG AGG AGG GG | Fig EV1 | Yes (2.65) |
| Seq580 | CGG GGG TGG AGG GGG GGG AG | Fig EV1 | Yes (2.80) |
| Seq583 | CGG GAA GGG GGG GCG GAG GG | Fig EV1 | Yes (2.40) |
| Basic Anti-Parallel | GGG GTT TTG GGG | Fig 3 | Yes ref: (Haider et al, 2002) |
| Thrombin Binding Aptamer (TBA) | GGT TGG TGT GGT TGG | Fig 3 | Yes ref: (Macaya et al, 1993) |
| Core Human Telomer Quadruplex | AGG TTA GGG TTA GGG TTA GGG | Fig 3 | Yes ref: (Renčiuk et al, 2009) |
| Wild Type c-MYC | TGA GGG TGG GGA GGG TGG GGA AGG | Fig 3 | Yes ref: (Simonsson et al, 1998; Phan et al, 2004; Ambrus et al, 2005) |
| MYC22 | TGA GGG TGG GGA GGG TGG GGA A | Fig 3 | Yes ref: (Phan et al, 2004; Ambrus et al, 2005) |
| MYC12 | TGG GGA GGG TTT TTA GGG TGG GGA | Fig 3 | Yes ref: (Phan et al, 2004) |
| PARP1 Promoter | TGG GGG CCG AGG CGG GGC TTG GG | Fig 3 | Yes ref: (Sengar et al, 2019) |
| LTR-III | GGG AGG CGT GGC CTG GGC GGG ACT GGG G | Fig 3 | Yes ref: (Butovskaya et al, 2018) |

For the full list of sequences used, see the supplemental data file.
[a]Lettering from left to right in figure.

1 μM DNA, which was the same concentration used in the holdase assays. The NMM spectra indicate that all three sequences form parallel quadruplexes at the concentration used in aggregation assays, unlike the ssDNA control (Fig 3B).

Circular dichroism melting experiments indicate that these quadruplex structures are stable. At 45°C, > 90% of the quadruplex structures remained for sequences 536 and 576, and 70% for sequence 359 (Fig EV2).

We also checked whether RNA versions of these sequences would also form quadruplexes. For 536 and 576, the conformation was similar for both DNA and RNA, with all spectra indicating parallel quadruplex formation. However, for sequence 359, the type and stability of quadruplex conformation depended on the type of nucleic acid present. The DNA version at 45°C had spectra showing a mixture of parallel and anti-parallel quadruplex. The RNA version appeared to be more stable, with 80% of the original quadruplex structure remaining at 45°C. Furthermore, the spectra indicate that only the parallel quadruplex is formed in the case of sequence 359 RNA. The change in structure for 359 from a mixed topology to parallel merited separate testing of its chaperone activity. Heat denaturation assays demonstrate that the RNA version also had potent chaperone activity, preventing aggregation even more than its DNA counterpart (Fig EV3).

These experiments confirmed that the holdase activity of these polyG-containing sequences is associated with quadruplex structure. Re-analyzing the heat denaturation aggregation assay data presented above, of the 160 sequences tested that had a polyG motif, 133 appeared in the top third of data, making up 79% of the sequences in that subset. 152 of the 160 polyG sequences also decreased aggregation by at least 50%.

The higher level of activity from quadruplex DNA raised the question of whether any structured DNA could have a similar effect. In other words, could the activity arise from any DNA with greater structure than ssDNA? To test this possibility, we tested the holdase activity of 24 duplexed sequences to compare directly with their single-stranded counterparts. Overall, the differences were small, and in many cases statistically insignificant (Fig EV4). These experiments suggest that the holdase activity displayed here could be specific to quadruplex structures, and not to other structured DNAs.

A related question was whether different forms of quadruplex structures have different intrinsic aggregation or chaperone tendencies. To test this question, we chose sequences that had previously determined quadruplex structures (Sundquist & Klug, 1989; Macaya *et al*, 1993; Simonsson *et al*, 1998; Haider *et al*, 2002; Phan *et al*, 2004; Ambrus *et al*, 2005; Renčiuk *et al*, 2009; Butovskaya *et al*, 2018; Sengar *et al*, 2019), and that were ~20 bases in length and subjected them to the heat denaturation aggregation experiment. The different sequences displayed varying activities that appeared to correlate with structural properties (Fig 3C). Namely, anti-parallel quadruplexes appeared to have no chaperone activity or appeared to increase protein aggregation, and parallel quadruplexes had minor chaperone activity, but 3 + 1 mixed quadruplexes displayed greater chaperone activity. These results suggest that the type of quadruplex matters in its chaperone function, along with adjacent sequences, consistent with the observations of our best tested sequence, 359.

## Generality of holdase activity

To determine the generality of this quadruplex holdase activity, we performed aggregation assays with three other proteins, luciferase, lactate dehydrogenase (LDH), and malate dehydrogenase (MDH). To check whether this holdase activity is quadruplex-specific with multiple proteins, we tested 14 sequences predicted to form quadruplexes by G4Hunter and 10 single-stranded sequences with each

protein (Table 1). These proteins have varying structural properties, ranging in pI from 6.1 to 8.5, and size from 62.9 kD to 140 kD. With all four proteins, the quadruplexes severely decreased protein aggregation, demonstrating strong holdase activity. For luciferase, LDH, and MDH, most of the quadruplex sequences tested were able to completely prevent protein aggregation. However, the single-stranded sequences demonstrated little to no activity for all of these proteins (Fig 4). These data strongly suggest that the holdase activity displayed by quadruplex sequences is general, while also unique to quadruplex-forming sequences. Of note, LDH is the only of these proteins with previously characterized DNA-binding activity toward both duplex and ssDNA (Cattaneo *et al*, 1985; Grosse *et al*, 1986; Fang *et al*, 2000), but its aggregation was only significantly reduced by binding to quadruplex sequences (Fig 4). Of note, two sequences (O and P) that G4Hunter did not predict as having high quadruplex probability but had significant holdase activity, do have substantial guanine content and could potentially still form quadruplexes despite being listed as ssDNA here.

## Chaperone activity in *Escherichia coli*

With our newfound quadruplex-containing sequences possessing powerful holdase activity *in vitro*, we sought to test whether they would have chaperone-like activity in a cellular system. For these experiments, we assayed the ability of these nucleic acids to improve the folding of fluorescent proteins in *E. coli*.

GFP's fluorescence is dependent almost solely on its folding to its native state, which allows self-catalyzed chromophore formation, and continued maintenance of the native state causes continued fluorescence (Craggs, 2009). Furthermore, although many variants of GFP were later engineered to have fast maturation times and high stability, wildtype GFP (wtGFP) folds slowly and poorly in *E. coli*, thereby producing little fluorescence. These properties previously enabled directed evolution of chaperones in *E. coli* to improve chaperone activity and therefore increase wtGFP fluorescence brightness (Wang *et al*, 2002). Similarly, wtGFP fluorescence can be used to monitor the protein folding stress level in *E. coli* (Song *et al*, 2016). Of importance, the underlying reason why expressing proteins with GFP fusions in *E. coli* increase their overall folding levels is due to productive folding interactions between the GFP and known chaperones (Zhang *et al*, 2005). We have confirmed the previous findings that co-expressing known chaperones increases the fluorescence (and therefore the folded protein) of wtGFP in *E. coli* (Fig EV5).

One major barrier existed to using wtGFP to examine chaperone nucleic acid activity. Both in our hands (Fig EV5) and others (Aarum *et al*, 2020), bulk nucleic acids have little to no effect on GFP folding or aggregation. This lack of effect likely stems from electrostatic inhibition, as wtGFP is quite acidic (pI = 5.67), and would be repelled by the negative charge of the nucleic acids' phosphate backbone. However, there are now many other GFP-like variants with differing sequence properties.

TagRFP675 was especially attractive due to its combination of its basicity (pI = 8.53), relatively long maturation time that would make it susceptible to aggregation or degradation, and long wavelength to allow easy detection (Piatkevich *et al*, 2013). Testing the same set of chaperones as wtGFP with TagRFP675 demonstrates that like wtGFP, the amount of folded TagRFP675 is considerably higher in *E. coli* co-expressing many known chaperones (compared

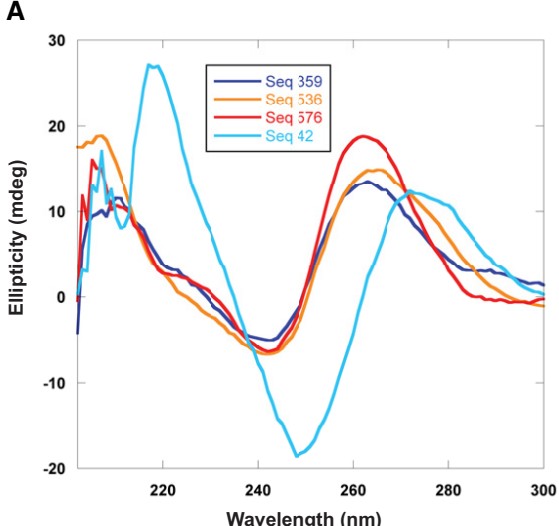

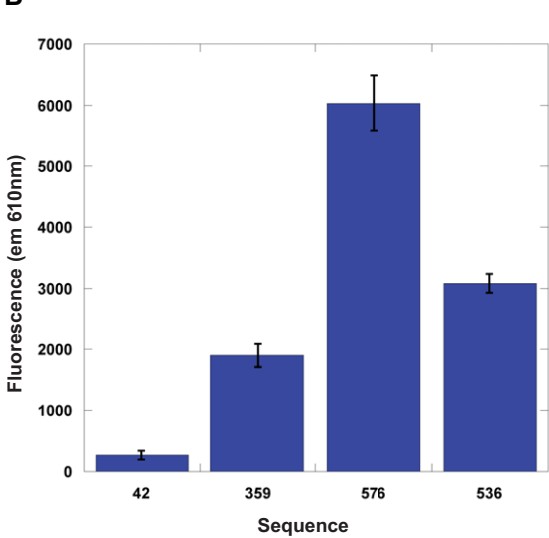

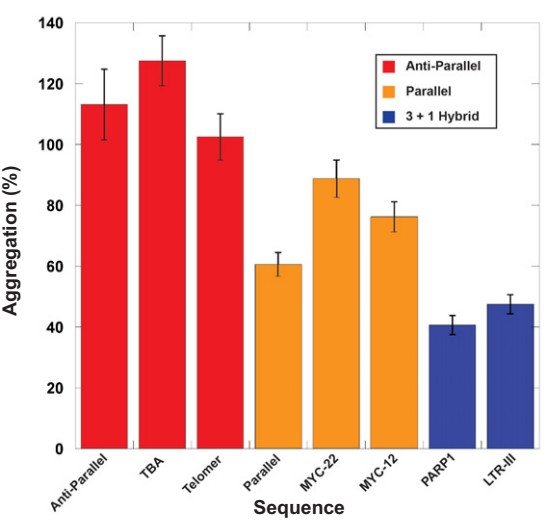

**Figure 3. Characterization of quadruplex content and holdase activity.**

A  Structural characterization of holdase nucleic acids using circular dichroism in sodium phosphate buffer. Peaks are observed at 260 nm and 210 nm, as well as a trough at 245 nm, indicating the presence of parallel G-quadruplexes. Thermal stability of quadruplexes shown in Fig EV2 in potassium phosphate buffer.

B  NMM fluorescence measured at 610 nm. Error bars are mean ± SE of technical triplicates.

C  Comparing holdase activity of different quadruplex-containing sequences of known topology (Sundquist & Klug, 1989; Macaya *et al*, 1993; Simonsson *et al*, 1998; Haider *et al*, 2002; Phan *et al*, 2004; Ambrus *et al*, 2005; Renčiuk *et al*, 2009; Butovskaya *et al*, 2018; Sengar *et al*, 2019). Error bars are mean ± SE of technical triplicate.

Source data are available online for this figure.

to the empty vector control) (Fig 5A–C). Notably, the only known chaperone to not cause an increase in the fluorescence in the TagRFP675 is Hsp33 (Fig 5B and C), which is activated specifically by oxidative stress (Winter *et al*, 2005, 2008), and would not be expected to be active under our conditions.

We then compared the fluorescence of TagRFP675 when co-expressed with our three best-characterized quadruplex-containing nucleic acids, sequences 359, 536, and 576, under heat stress. As controls, we compared both against empty vector, as well as sequence 42, which displayed little to no *in vitro* activity. Expressing the quadruplex-containing or control sequences did not significantly affect *E. coli* growth rate (Appendix Fig S1). Of note, unlike our *in vitro* testing, in this experiment, the quadruplex-containing sequences are expressed as RNA from a pBAD33 plasmid modified for small RNA expression. Notably, this plasmid does not contain a ribosome-binding site, and so little translation of these RNAs is expected.

Testing the fluorescence of TagRFP675 in each of these cases demonstrates that there is considerably higher fluorescence in cells co-expressing the quadruplex-containing sequences than either the empty vector or Seq42, indicating a higher level of properly folded TagRFP675 (Fig 5B and C). The effects of the quadruplex-containing sequences were greater than that of several known chaperones, including DnaK, Hsp33, IbpA, and IbpB (Fig 5B and C). As a control, we also tested the acidic wtGFP, which was unaffected by the quadruplex-containing sequences (Fig EV5). These experiments showed that the chaperone nucleic acids found *in vitro* are also able to improve the folding of a fluorescence protein that usually struggles to stably fold in *E. coli*. Of note, we have only tested three of the quadruplex-containing sequences chosen from our *in vitro* screen, and all three have shown chaperone-like effects. The three chaperone sequences we have tested thus far ranked 1st, 18th, and 52nd in our initial screen; based on this ranking we would also expect dozens more sequences from our screen to potentially display chaperone activity in *E. coli*.

## Holdase activity due to oligomerization

While analyzing the light scattering data from chemically denatured citrate synthase in the presence of quadruplexes, we noticed that although the total light scattering was greatly decreased by the presence of the quadruplexes, the quadruplexes caused a small initial

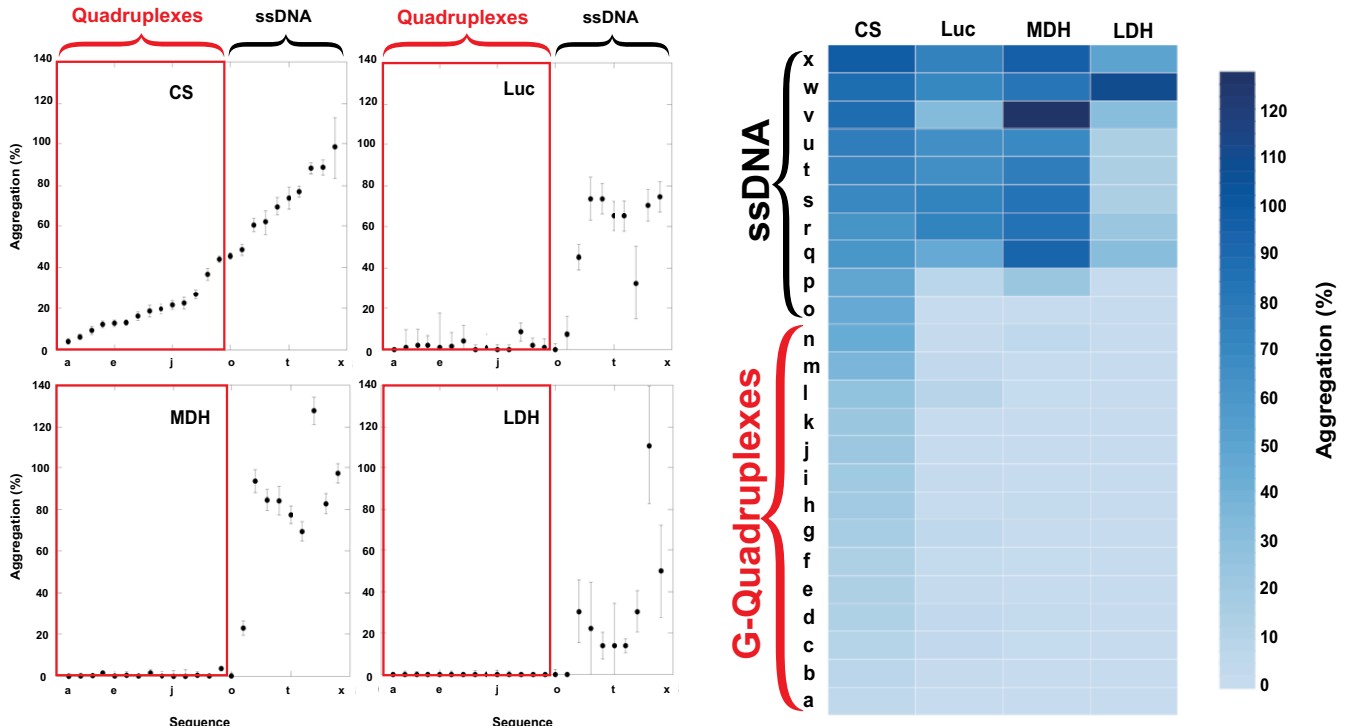

**Figure 4. Generality of G-quadruplex holdase activity using four different proteins.**

Luciferase (Luc), Citrate Synthase (CS), L-Malate Dehydrogenase (MDH), and L-Lactate Dehydrogenase (LDH) boxed in red are the 14 sequences with the propensity to form quadruplexes (as identified by G4Hunter), while the remaining 10 sequences are non-structured ssDNA (left). These data were also shown as a heat map (right). Error bars are mean ± SE of technical triplicates.

Source data are available online for this figure.

jump in light scattering (Fig 6A). These data are highly reminiscent of the pattern we observed recently in which nucleic acids could prevent protein aggregation by promoting protein:nucleic acid oligomerization (Litberg *et al*, 2019). In this previous study, we found that bulk nucleic acids at high concentration could prevent protein aggregation under extreme conditions through the formation of protein:nucleic acid oligomers that could be controlled by varying nucleic acid concentration (Litberg *et al*, 2019). While the best quadruplex-containing sequences were considerably more efficient than the bulk DNA used previously, the initial jump in light scattering hinted that the initial holdase assays could have provided an overly simplistic interpretation of their mechanism, and that oligomerization could be at least partially responsible for their activity.

To determine whether the quadruplexes tested here were acting in a similar manner, we performed additional spin-down assays, CD, and transmission electron microscopy (TEM) experiments. Spin-down assays were performed by heating citrate synthase, luciferase, MDH, or LDH with quadruplexes at 60°C for 15 min, and then returning them to room temperature. The sample was centrifuged to then separate soluble and pellet fractions. SDS–PAGE gels demonstrated that the quadruplexes kept the proteins soluble even at extreme temperatures (Fig 6B), similar to previously characterized oligomerization cases (Litberg *et al*, 2019). Of note, a control single-stranded sequence of the same length with little chaperone activity in our previous assays (sequence 42) did not keep the

proteins soluble under these conditions, even for the well-characterized DNA binding protein LDH (Fig 6B). Measuring CD spectra of luciferase protein as a function of temperature in the presence of quadruplexes showed that the protein maintained partial β-sheet structure as high as 80°C, and that this non-native structure was retained upon return to room temperature (Appendix Fig S2), similar to previous oligomerization cases (Litberg *et al*, 2019). Finally, negative stain TEM imaging showed that the quadruplexes caused the formation of small protein oligomers (Fig 6C). The morphology of these oligomers was similar to previously observed oligomers formed in the presence of bulk DNA, suggesting similar mechanisms to our previous observations, in which nucleic acid:protein oligomers are kinetically stable (Litberg *et al*, 2019). Further spin-down assays in chemically denaturing conditions suggest that these oligomers tend to not form large aggregates (Appendix Fig S3).

## Discussion

In this study, a systematic investigation of the holdase activity of nucleic acids demonstrated that this activity is sequence specific, and that quadruplex sequences display potent holdase activity. This chaperone activity also depends on the quadruplex topology. This holdase activity was demonstrated to be general, preventing the aggregation of multiple proteins that differed considerably in pI,

**A**

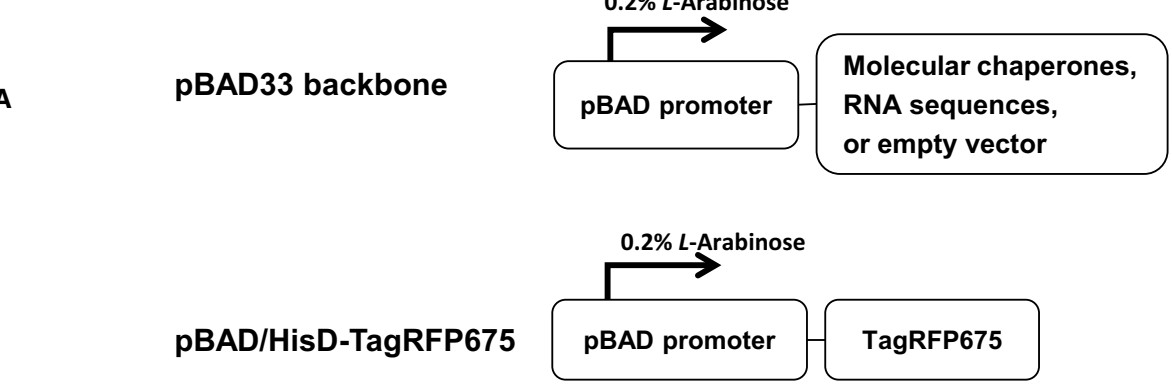

**B**

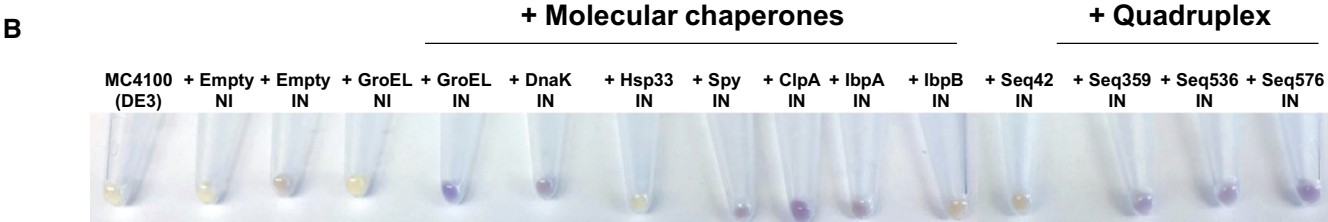

**C**

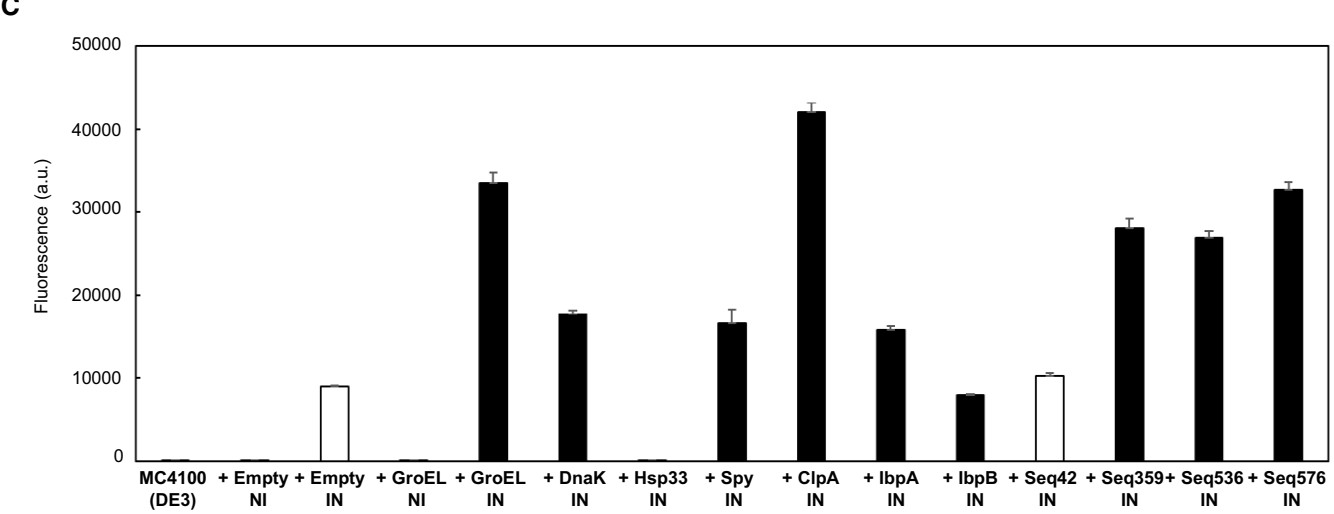

**Figure 5.** **G-quadruplex-containing sequences enhance the fluorescence of the biosensor protein at 42°C.**

A  A schematic illustration of expression vectors. Both the expression of protein folding enhancing factors and TagRFP675 are under the control of pBAD promoter, which is induced by 0.2% L-Arabinose.

B  Harvested cells. Empty vector (negative control), molecular chaperones (GroEL, DnaK, Hsp33, Spy, ClpA, IbpA, and IbpB), and selected RNA sequences (Seq42, Seq359, Seq536, and Seq576) were used as protein folding enhancers. NI and IN indicate: non-induced and induced, respectively.

C  Cellular fluorescence assay of TagRFP675 with various protein folding enhancing factors. Protein expression was induced at 42°C, and the fluorescence of each sample was measured with spectrophotometer. White bars (+Empty IN and + Seq42) indicate negative controls. The data are shown as mean ± SD of technical triplicates ($n = 3$).

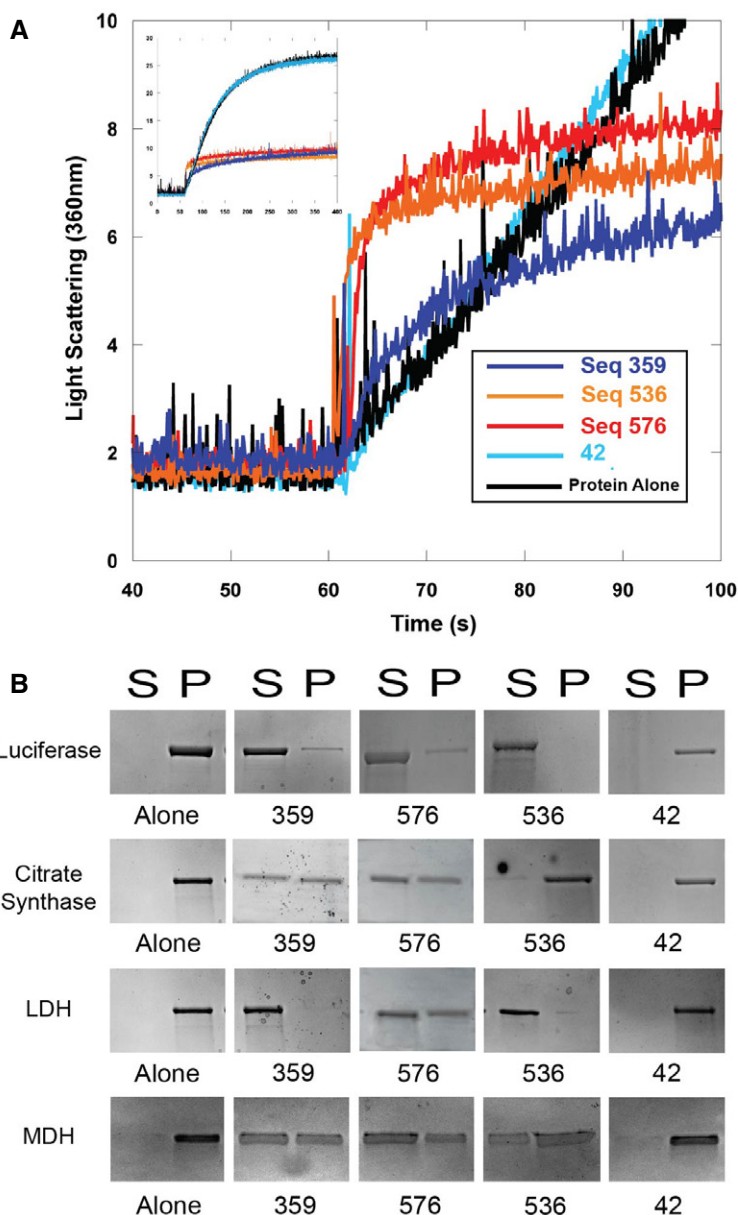

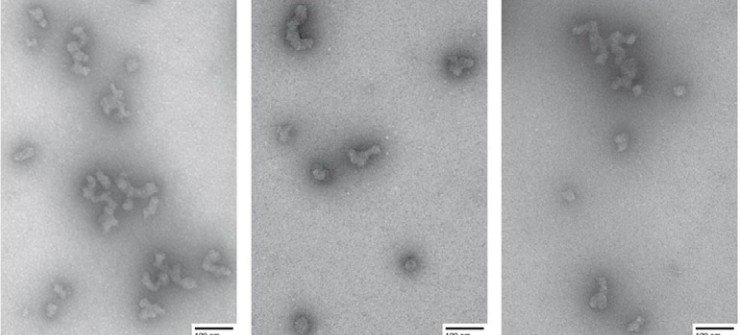

**Figure 6.**

**Figure 6. Quadruplex-containing sequences promote oligomerization.**

Sequences 359, 536, and 576 all displayed holdase activity and contain a polyG motif. Sequence 42 was used as a negative control, as it performed poorly as a holdase chaperone and did not contain a polyG motif.

A  Right angle light scattering of chemically induced aggregation of citrate synthase. Initial kinetics of aggregation shown with full time scale as the insert.

B  Spin-down assay with four different proteins denatured at 60°C in the presence of DNA. S represents the soluble fraction while the P represents the insoluble fraction or pellet.

C  Transmission electron microscopy negative stain images of soluble fractions from thermally induced aggregation spin-down assays. Citrate synthase oligomers were observed in each of the quadruplex cases, although the morphology of the quadruplex was dependent on the DNA sequence. Corresponding thermal denaturation spin-down assays are shown in panel (B). Scale bar is 100 nm.

size, and function. Further testing showed that this activity arises largely via protein:nucleic acid oligomerization. To our knowledge, this activity was also found to be more efficient than any previously characterized protein chaperone (Docter *et al*, 2016).

Despite the ability of nucleic acids to serve as powerful chaperones *in vitro* (Docter *et al*, 2016; Litberg *et al*, 2019), to our knowledge, chaperone-like activities described in living cells for nucleic acids are restricted to well-known RNA binding proteins with typical RNA-binding motifs (Mann *et al*, 2019), or for RNAs binding to their evolved RNA-binding partners (Son *et al*, 2015). In this work, we describe the ability of nucleic acids optimized for chaperone activity *in vitro* to improve the folding environment for a separate protein in *E. coli*, with the activity similar to that of many known chaperones. Based on the differences between wtGFP and TagRFP675, it appears that electrostatics are an important component of this activity in *E. coli*. Our findings on quadruplex-containing sequences are consistent with recent biological observations. It was recently discovered that highly structured RNAs can interact with many proteins that do not contain typical RNA-binding domains (Sanchez de Groot *et al*, 2019). Also of note, in addition to maintaining the fluidity of stress granules (Maharana *et al*, 2018), RNAs can bind to proteins without typical RNA-binding activity, and escort them to stress granules (Bounedjah *et al*, 2014; Alriquet *et al*, 2019). Of especial interest, a recent study found that in response to oxidative stress, mitochondrial quadruplex DNA is excised from the mitochondrial genome, and migrates to the cytoplasm, where it specifically helps form stress granules (Byrd *et al*, 2016). It is highly likely that the chaperone and oligomerization behavior observed here for quadruplex-containing sequences underlies this capacity to accumulate in and help form stress granules. However, it is worth bearing in mind that quadruplexes are often localized utilizing the BG4 antibody, which has recently had its specificity called into question (Ray *et al*, 2020). Similarly, our work here was in *E. coli*, in which quadruplexes form readily, but form less readily in eukaryotes (Guo & Bartel, 2016).

Quadruplex sequences have recently been implicated in aggregation and phase separation events in the cell that are associated with pathology. The quadruplex-forming GGGGCC repeat expansion in the c9orf72 gene is thought to be a frequent cause of both ALS and frontotemporal dementia (FTD; DeJesus-Hernandez *et al*, 2011; Renton *et al*, 2011). This quadruplex sequence is transcribed into sense and anti-sense RNAs that have been shown to sequester numerous RNA binding proteins into toxic intranuclear foci, resulting in aggregation (Donnelly *et al*, 2013; Haeusler *et al*, 2014; Conlon *et al*, 2016; Jain & Vale, 2017; Simone *et al*, 2018). GGGGCC quadruplexes forming foci with disordered proteins in the cell are highly consistent with the results presented here. FMRP protein has been shown to be one of the leading causes of the fragile X

syndrome (Zhang *et al*, 2014) as well as one of the leading causes of monogenetic forms of autism (Bassell & Warren, 2008; Hernandez *et al*, 2009). In addition to aggregating in diseases (Sjekloća *et al*, 2011; Derbis *et al*, 2018), FMRP is a known quadruplex-binding protein (Zhang *et al*, 2014; Vasilyev *et al*, 2015). Future studies on whether these roles are related would be of significant interest. The results presented here suggest that these disease-relevant cases are not unique, as this behavior is a general property of quadruplex interaction with partially unfolded or disordered proteins under stress conditions. Our topology studies suggest that the parallel vs anti-parallel nature of these systems could have a strong effect on their oligomerization and aggregation.

It is also interesting to note that the quadruplex sequences with highest activity also contain sequence regions that are not expected to form quadruplexes or regular secondary structure, suggesting that the inter-relation of the quadruplex and single-stranded regions could be important for activity. Indeed, our previous investigations suggested that pure polyG by itself is not a very efficient chaperone. The sequences characterized here to have high activity likely bear some structural resemblance to known chaperones; specifically, the end of the quadruplex is a large hydrophobic surface, surrounded on four sides by flexible charged tails of single-stranded sequences. This arrangement is very similar to the important features and structure of the chaperone Spy (Horowitz *et al*, 2016). While there are very likely differences in activity as well, this similarity suggests that we could use known chaperone mechanisms as a starting place for understanding the chaperone activity of quadruplex-containing sequences. Alternatively, testing different known topologies of quadruplexes showed very different aggregation properties. Specifically, the large difference in activity between anti-parallel and mixed 3 + 1 topology suggests that the quadruplex topology is an important component in its oligomerization properties. Despite the number of sequences tested here, many quadruplex variants remain for further future exploration. Similarly, the tests here were performed with a single length of nucleic acid (20 bases), whereas greater topology variance could likely be achieved with longer sequences. Future work could greatly refine the proof-of-principle concepts investigated here.

Quadruplexes preventing protein aggregation by oligomerizing with their clients is reminiscent of the action by small heat shock proteins (sHsps). This class of chaperones forms large hetero-oligomer complexes with different clients, keeping these clients soluble and in an accessible state for later refolding by ATP-dependent chaperones (Jakob *et al*, 1993; Haslbeck & Vierling, 2015). The quadruplexes studied here also formed stable and soluble oligomer complexes with partially folded proteins, suggesting that the quadruplexes could use a molecular mechanism similar to those of the sHsps. Of note, the morphology of the oligomers varied with

quadruplex sequence, further suggesting that altering quadruplex sequence could be a way to control the oligomerization of proteins under stress conditions.

# Materials and Methods

### Sourcing nucleic acids

All were ordered from Integrated DNA Technology using their standard desalting and purification procedures. For the heat aggregation plate reader assays, DNA was ordered lyophilized, and normalized to guaranteed molar weights by IDT. This DNA was then resuspended in the given buffer and pipetted directly into the plate wells after thorough pipette mixing. The duplex DNA was pre-annealed by IDT using their standard annealing protocol (https://www.idtdna.com/pages/education/decoded/article/annealing-oligonucleotides). For all other experiments, DNA or RNA was ordered lyophilized in tube form from IDT at the maximum yield achieved during synthesis. The nucleic acids were then resuspended and thoroughly mixed with the given buffer or DEPC-treated water to a known concentration.

### Thermal aggregation plate reader assays

For the initial thermal aggregation assays, 312 single-stranded sequences of random sequence, 24 of which varied in length from 15 to 20 bases long, while the rest were all 20 bases long, were incubated with 500 nM Citrate Synthase from porcine heart (Sigma-Aldrich C3260-5KU) in a 1:2 protein:DNA strand concentration. Aggregation was measured absorbance at 360 nm in a Biotek Powerwave multimode plate reader, using black clear flat bottom half-area plates (Corning 3880), with shaking and measurements every 36 s. In every assay, the plates were transferred from ice to a preheated 50°C plate reader, and the temperature was held constant throughout the entire experiment. Each plate was run for 1.5 h in 40 mM HEPES, pH 7.5 (KOH) buffer. The sequences were run in triplicate. Percent aggregation was calculated as a function of the maximum absorbance value recorded in the hour and a half divided by the maximum protein alone absorbance value. Error bars shown are standard error propagated from both the triplicate protein alone and triplicate experimental measurement. As a control, herring testes DNA (Sigma) was also run on each plate to ensure consistency of data.

The enriched population of 192 G-rich sequences (length 20 bases) was performed by biasing 96 of the sequences toward guanine bases at a rate of 55%. 40 sequences were biased toward guanine bases by 75%, and the remaining 56 sequences were created with the motif GGGGGNT systematically placed throughout the sequence, with the remaining 13 bases chosen at random. This process was accomplished by altering the random bias in our random sequence generating software, which can be found at: https://github.com/adambegeman/IDT_DNA_Generator.

The heat denaturation assay with varying DNA concentration was run identically to that described above, with ssDNA strand:protein ratios of: 0.5:1, 1:1, 2:1, 4:1, and 8:1. The known quadruplex heat denaturation assay was also run identically to that described above, with ssDNA of known quadruplex-forming sequences

(Table 1). The sequence 359 RNA heat denaturation assay used 0.1% DEPC-treated autoclaved and filtered DI water.

We also tested aggregation using Quantilum Recombinant Luciferase (Promega), *L*-malate dehydrogenase (MDH) from pig heart (Sigma-Aldrich 10127914001), and *L*-Lactate Dehydrogenase (LDH) from rabbit muscle (Sigma-Aldrich 10127876001). We chose 24 sequences from the citrate synthase assay whose anti-aggregation ability spanned the entire range of our data. Of these 24, 14 had a propensity to form quadruplexes. The assay with these other three proteins was carried out identically to the citrate synthase thermal denaturation assay, with LDH being run for 3 h due to its higher stability at 50°C. These assays were run in 1:2 protein:DNA strand ratios using 10 mM sodium phosphate, pH 7.5 buffer at protein concentrations of 500 nM luciferase, 2 μM MDH, and 4 μM LDH.

### Motif analysis

Motif analysis was performed using the HOMER package (Benner *et al*, 2017) to compare the one-third of sequences with the highest holdase activity to the one-third of sequences with the lowest holdase activity. The parameters used were "-len 5,6,7,8,9,10 -norevopp -noconvert -nomask -mis 2 -basic -nogo -noredun -noweight -fdr 1000". We also used MatrixREDUCE (Foat *et al*, 2006) to identify motifs correlated with holdase activity by regression method. Spearman correlation analysis was performed by R.

### Chemical aggregation light scattering

Procedure was adapted from Docter *et al* (2016) and Gray *et al* (2014). For the chemically induced aggregation, 12 μM Citrate Synthase was denatured in 6 M guanidine-HCl, 40 mM HEPES, for approximately 16 h at 23°C, then diluted to 75 nM into 40 mM HEPES, pH 7.5 (KOH), with constant stirring at 23°C in the presence of 150 nM 20-mer ssDNA. The resulting aggregation was then measured via right angle light scattering at 360 nm in a fluorimeter with constant mixing. Results were consistently repeated on three separate days, with representative curves shown.

### N-methylmesoporphyrin IX fluorescence

N-methylmesoporphyrin IX (NMM) is a well-characterized fluorophore that increases fluorescence when bound to parallel quadruplexes (Sabharwal *et al*, 2014). The emission spectra of 10 μM NMM was measured using an excitation wavelength of 399 nm, and an emission range of 550–750 nm in the presence of 1 μM DNA in 10 mM sodium phosphate, pH 7.5. Samples were run in triplicate at 25°C in a multimode plate reader. Reported values are taken at 610 nm, the emission maxima, as a function of increase in fluorescence compared to an NMM alone triplicate control.

### Construction of expression vectors

Expression vectors were generated of protein folding enhancing factors, i.e., pBAD33-molecular chaperone (GroEL, DnaK, Hsp33, ClpA, Spy, IbpA, and IbpB), pBAD33mut-RNA (Seq42, Seq359, Seq536, and Seq576), and pBAD33mut-Empty. The expression vectors of pBAD33-molecular chaperone (GroEL, DnaK, Hsp33, ClpA, Spy, IbpA, and IbpB) were constructed from pBAD33 using

the SacI and HindIII sites. Each molecular chaperone gene was obtained by PCR using *E. coli* genome. pBAD33mut-empty vector (Empty) was generated from pBAD33 vector. Briefly, the MCS of the pBAD33 vector was changed into SacI and SpeI, using MluI and BglII restriction enzymes. The genes of the selected set of RNAs (Seq42, Seq359, Seq536, and Seq576) were synthesized (Genscript). pBAD33mut-Seq42, Seq359, Seq536, and Seq576 were generated using the SacI and SpeI sites. For the expression of biosensors, we used pBAD/HisD-TagRFP675 and pBAD18-wtGFP. pBAD/HisD-TagRFP675 was a gift from Vladislav Verkhusha (Addgene plasmid # 44274; http://n2t.net/addgene:44274; RRID:Addgene_44274) (Piatkevich *et al*, 2013). pBAD18-wtGFP was given by Jonathan S. Weismann' laboratory (Wang *et al*, 2002). Strains and plasmids used are listed in Table EV1.

## Protein expression, and *Escherichia coli* fluorescence assay

Each resulting expression vector of protein folding enhancing factors and pBAD/HisD-TagRFP675 or pBAD33-wtGFP were co-transformed into the *E. coli* strain MC4100(DE3) by heat shock. To induce protein and/or RNA expression, each transformant was streaked on an LB plate containing 0.2% *L*-Arabinose, ampicillin (200 μg/ml), and chloramphenicol (50 μg/ml) at 42°C overnight. To confirm whether pBAD promoter is tightly controlled, non-induced LB plates containing the same amount of antibiotics were used for both +Empty and +GroEL samples. The next day, non-induced and induced cells were scraped in 170 mM NaCl and harvested by centrifugation at 10,000 *g* for 2 min at 4°C. Pictures of cell pellets were taken immediately after harvesting to compare color. Cells were then resuspended in 170 mM NaCl and diluted to 0.1 OD$_{600}$ for further fluorescence assays. Fluorescence emission of each sample was measured by microplate reader (Infinite M200 Pro, Tecan) using a black 96-well plate (Corning Black NBS 3991). Fluorescence emissions of TagRFP675 (at 576 nm) and wtGFP (at 395 nm) were measured upon excitation at 598 nm and 509 nm, respectively.

## *Escherichia coli* growth assay

To measure cell growth of each sample, each strain was inoculated into 3 ml of LB medium containing ampicillin (200 μg/ml) and chloramphenicol (50 μg/ml), and cultured at 37°C overnight. Next day, 1 μl of each culture was transferred into 99 μl of LB medium containing 0.08% *L*-Arabinose, ampicillin (200 μg/ml), and chloramphenicol (50 μg/ml). The samples were added into a clear flat bottom 96-well plate (Corning CLS3997) and grown for 19 h at 42°C with shaking for 10 min every hour. OD$_{600}$ of each sample was measured with a microplate reader (Infinite M200 Pro, Tecan) every hour. All samples were measured in triplicate.

## GFP *in vitro* fluorescence assay

A GFP stock was prepared using Recombinant *A. victoria* green fluorescent protein (Abcam ab84191) in 20 mM potassium chloride and 20 mM Tris, pH 7.5 buffer (HCl) containing 1 mM DTT. Herring Testes DNA (htDNA) was prepared using deoxyribonucleic acid sodium salt from herring testes (Sigma) dissolved in 20 mM KCl and 20 mM Tris, pH 7.5 buffer. The htDNA solution was washed three times at 4°C, using a 3 kD molecular cutoff weight cellulose

ultra-centrifuge spin filter (Sigma Amicon), according to manufacturer's protocol. All samples were prepared on ice.

Experimental samples were prepared with a final GFP concentration of 1.4 nM containing 0.2408 mg/ml htDNA or GFP alone, as a control. Samples were then denatured at 93°C for 20 min. Following denaturing, 100 μl of each sample was immediately transferred to a treated black 96-well plate (Corning Black NBS 3991). Fluorescence was measured at 37°C using a fluorescent plate reader (excitation/emission: 390/508 nm), over a 60-minute incubation period. All samples were blank subtracted from buffer containing 0.2408 mg/ml htDNA. Experiments were controlled for using non-denatured GFP with htDNA, at the same concentrations. All samples were measured in at least triplicate.

## Transmission electron microscopy (TEM)

For the oligomer TEM samples, thermal denaturation spin-down assays were run using citrate synthase. 100 μl of 3.8 μM protein and 7.6 μM ssDNA were thermally denatured together at 60°C for 15 min in 10 mM sodium phosphate, pH 7.5 buffer. The resulting solution was then centrifuged at 16,100 *g* for 15 min at 4°C to separate the soluble and insoluble fractions. The soluble portion was pipetted off and transferred on ice for TEM analysis.

A positively charged copper mesh grid coated in formvar and carbon (Electron Microscopy Sciences) using the PELCO easiGlow Discharge system was used for each soluble sample. The charged copper grids had 5 μl of sample applied for 20 s and then lightly blotted off using a Whatman filter paper. The grids were then rinsed using 2 drops of MilliQ water, with filter paper blotting for each wash. Finally, the grids were then stained using two drops of a 0.75% uranyl formate solution. The first drop served as a quick wash, followed by 20 s of staining using the second drop. The grids were then blotted and allowed to dry. The TEM images were captured using a FEI Tecnai G2 Biotwin TEM at 80 kV with an AMT side-mount digital camera. In order to better visualize the intricacies of each oligomer, the images' contrast and brightness was uniformly enhanced using Adobe Photoshop.

## Circular dichroism

All CD spectra were obtained using a Jasco J-1100 circular dichroism at 23°C.

For CD spectra of DNA in Fig 3A, the sequences were resuspended in 10 mM sodium phosphate pH 7.5 buffer and diluted in the same buffer to 25 μM (per strand) DNA. The CD measurements were taken from 300 nm to 190 nm at 1 nm intervals using a 50 nm/min scanning speed. The shown spectra are a product of three accumulations using the same conditions.

The luciferase protein denaturation CD spectra were captured in protein:DNA ratios of 1:2 using 3.2 μM luciferase. The CD spectra were captured at 10°C intervals from 15°C to 85°C using a ramp rate of 3°C/min captured from 260 to 190 nm. The concentrations were chosen such that the DNA concentration was below the observable sensitivity range of the instrument.

For nucleic acid melting curves, the sequences were resuspended in 10 mM pH 7.5 potassium phosphate for DNA or DEPC-treated DI water for RNA, both of which were diluted into 10 mM potassium phosphate for spectra acquisition. Spectra were taken at 10°C

intervals from 25°C to 85°C and back to 25°C at a ramp rate of 2°C/min. Spectra were captured from 320 nm to 190 nm at 1 nm intervals using a 50 nm/min scanning speed over three accumulations.

For annealed quadruplex sequence spectra, quadruplex-forming sequences were heated to 95°C for two minutes and allowed to cool to room temperature by the internal fan of the heating block, which took 30 min to cool to 25°C. Samples had their CD spectra taken immediately upon their return to 25°C.

**Spin-down aggregation assays**

For the aggregation assays, 100 μl of 3.2 μM protein and 6.4 μM ssDNA were thermally denatured together at 60°C for 15 min in 10 mM sodium phosphate, pH 7.5 buffer. The resulting solution was then centrifuged at 16,100 *g* for 15 min at 4°C to separate the soluble and insoluble fractions. After centrifugation, the supernatant (approximately 97 μl) was removed and the pellet resuspended using 1 mM β-mercaptoethanol (Fisher Scientific) in 1× TG-SDS buffer (Bio Basic Inc) to the original sample volume of 100 μl. 10 μl of the soluble and pellet fractions were then run on a denaturing SDS–PAGE gel and visualized using Coomassie blue. Gels were reproduced on three separate days, with representative assays shown.

For the chemical denaturation spin-down assay, 46.4 μM citrate synthase was denatured in 4.8 M guanidine-HCl, 40 mM HEPES buffer for 16 h. The citrate synthase was then diluted to 2.5 μM in a 100 μl sample containing 5 μM of the target sequence that induced oligomerization. After 5 min, the samples soluble and pellet fractions were separated via the same methods as the thermal denaturation spin-down assays. The gels were then run in triplicate identically to the thermal denaturation experiments, with representative gels shown.

# Data availability

This study includes no data deposited in external repositories.

**Expanded View** for this article is available online.

## Acknowledgments
The authors would like to acknowledge E. Chapman, J. Yesselman, and S. Barbee for helpful conversation, and J. Weissman for the GFP vector. This work was supported by NIH R00 GM120388.

## Author contributions
SH conceptualized project. SH and AS administered project. SH, AB, TJL, and AS designed experiments assisted by all authors. ZX designed bioinformatics analysis. All authors analyzed data. AB, TJL, TG, VHC, THW, AS, JB, and ZX performed investigations. SH, AB, and AS wrote the paper with assistance from all authors.

## Conflict of interest
The authors declare that they have no conflict of interest.

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
