## [Review Process File · EMBO Reports]

G-Quadruplexes Act as Sequence-Dependent Protein Chaperones

Adam Begeman, Ahyun Son, Theodore Litberg, Tadeusz Wroblewski, Thane Gehring, Veronica Huizar Cabral, Jennifer Bourne, Zhenyu Xuan, and Scott Horowitz

DOI: [10.15252/embr.201949735](https://doi.org/10.15252/embr.201949735)

Corresponding author(s): Scott Horowitz (scott.horowitz@du.edu)

Review Timeline:

Submission Date:	22nd Nov 19
Editorial Decision:	10th Jan 20
Revision Received:	28th Feb 20
Editorial Decision:	31st May 20
Revision Received:	6th Jul 20
Editorial Decision:	24th Jul 20
Revision Received:	29th Jul 20
Accepted:	5th Aug 20

Editor: Esther Schnapp

Transaction Report:

Dear Scott,

Thank you for your patience while your manuscript was peer-reviewed at EMBO reports. It was sent to three referees, and so far we have received reports from two of them, which I copy below. Although the third referee has not returned his/her report, the other two referees are in fair agreement, so I am making a decision on your manuscript now in order to save you from any unnecessary loss of time.

As you will see, while the referees acknowledge that the observations are potentially interesting, they also both point out that the data as they stand are not particularly novel, and that the physiological relevance remains to be determined. Both referees also rate the technical quality of the study "low/unacceptable" in the manuscript summary table that is directly sent to the editor.

Given these substantial concerns, the amount of work required to address them, and the fact that EMBO reports can only invite revision of papers that receive enthusiastic support from the referees, I am sorry to say that we cannot offer to publish your manuscript.

That said, your work might be a good candidate for our partner journal Life Science Alliance (<http://www.life-science-alliance.org/>; our broad scope Open Access journal published in partnership between the EMBO-, Rockefeller University-, and Cold Spring Harbor Laboratory Presses). I have not discussed your work with the editors of Life Science Alliance, but you can easily transfer your manuscript by following the link below. Manuscript handling, peer review and publication is really swift at LSA. LSA executive editor Andrea Leibfried (a.leibfried@life-science-alliance.org) will be pleased to answer any questions and will work with the existing referee reports, if you agree.

For EMBO reports, I am sorry that I cannot be more positive this time. I nevertheless hope that the referee comments will be helpful in your continued work in this area.

Kind regards,
Esther

Referee #1:

In this paper, Begeman perform in vitro biochemical experiments to assess a range of nucleic acid sequences' effects on the aggregation of model protein substrates. In general, these studies are well conducted and controlled. However, many of the results presented here are not surprising. Prior work from the Bardwell lab (on which Horowitz is a coauthor) has already shown that that DNA and RNA exhibit potent chaperone activity in vitro. Although it is claimed that the sequence-dependence to this activity has been discovered (e.g. G-quads have high activity), whether the mechanisms found in vitro are also active in vivo remains undetermined. Indeed, it remains uncertain whether these sequences would act as chaperones in cells. Without in vivo evidence to support the in vitro data, my enthusiasm for this paper is very low. Moreover, the mechanism of anti-aggregation is proposed to be protein:nucleic acid oligomerization. Given that soluble, misfolded protein

oligomers can be highly damaging species in neurodegenerative disease, it could very well be that these species are cytotoxic and that this nucleic-acid chaperone activity is counterproductive in cells. However, this possibility is not studied, which further weakens this paper. Indeed, it is not clear that holding proteins in a misfolded oligomeric state is any better for cells than formation of aggregates when in many cases protein aggregates can be cytoprotective. Overall, the lack of in vivo studies to corroborate the in vitro studies here severely weaken this paper, which would be better suited to a more specialized biochemistry journal.

Referee #3:

The authors investigate the "holdase" activity (i.e. ability to prevent protein aggregation) of different DNA oligonucleotides (< 20 nt in length) in various protein aggregation assays. They find that oligos with sequences having a propensity to form G-quadruplexes tend to have the highest apparent holdase activity. The observations are interesting, but I have several concerns about the findings and presentation, and the biological significance of the findings seems quite unclear. My specific concerns are as follows.

1) A major problem with the manuscript is that there is no effort made to address the biological significance of the findings. First, the authors provide no discussion, not even a purely theoretical one, of how biologically plausible it is that G-quadruplexes within cells could have the "holdase" activity demonstrated in vitro. Second, and more importantly, for a high impact journal like EMBO Reports, my feeling is that some sort of direct biological evidence for the importance of the activity is essential. I list additional concerns below, but regardless of whether they are addressed or not, I think this principal concern is of central importance.

2) The description of the G-quadruplex-forming oligos needs to be improved. In particular, the authors indicate that these oligos contain the consensus GGGGGNT (and another similar one), but if they were to contain only one of such motif, this would require the presence of at least two such oligos in any G-quadruplex that is formed, and it isn't clear that such G-quadruplexes would form efficiently under the conditions employed (e.g. to form dimeric or tetrameric structures requires sufficiently high oligo concentrations and time). Fortunately, visual inspection of the sequences of the oligos in the supplemental material makes it very clear that most of them do indeed have a high capacity to form intramolecular G-quadruplexes (i.e. each oligo can form its own G-quad, most of them with stacks of only two guanine quartets, although in several cases dimers or tetramers would need to form), and so, together with the evidence in Fig. 2A and B, I do believe that many/most of the oligos in question do have G-quadruplex forming potential. I suggest that the authors: a) describe the actual oligo sequences in the main text so that readers are not confused about this issue, b) describe their annealing conditions for their CD experiments properly so that readers familiar with G-quadruplex biochemistry will be able to understand the findings in 2A, and c) use one of the various algorithms that are available (e.g. see Nucleic Acids Res. 2016 Feb 29; 44(4): 1746-1759) so as to more completely characterize and describe the G-quadruplex forming potential of the oligos they have found.

3) It's interesting that the 3 oligos that were characterized all appear to form parallel G-quadruplexes, but it's hard to know if this pattern is by chance given the small number of oligos tested. It would be interesting to test systematically the relative holdase activities of oligos that form parallel vs anti-parallel vs mixed (e.g. 3+1 hybrid) G-quadruplex folds, and to test whether G-quadruplexes with different numbers of stacked quartets (at least 2 vs 3) have different activities.

4) The holdase assays seem perhaps more complex than implied, and I don't feel that enough information is provided to understand how the oligos might be exerting their effects. For example, for the CS aggregation assay used for the screen, what does it mean that one oligo allows 80% aggregation and another allows 20%? - e.g. what is the dose-response relationship for oligo concentration vs holdase activity? It's hard to understand how much better the G-quad forming oligos are than non-G-quad without such context. Moreover, the authors describe holdase activity as "preventing protein aggregation", but doesn't the promotion of protein oligomerization (Fig 4) by the G-quads argue that it isn't that simple? It seems to me that the G-quads actually promote small aggregates to form --- i.e. they promote aggregates that are large enough to cause a small increase light scattering, but prevent further growth into aggregates that are as large as the aggregates formed in their absence (and so they scatter light less and are not spun down easily). By the way, how large are these small aggregates (the text under the scale bars in Fig 4C is too small to see), and what might this say about what might be happening in vivo?

** As a service to authors, EMBO Press provides authors with the ability to transfer a manuscript that one journal cannot offer to publish to another journal, without the author having to upload the manuscript data again. To transfer your manuscript to another EMBO Press journal using this service, please click on
Link Not Available

Referee #1:

In this paper, Begeman perform in vitro biochemical experiments to assess a range of nucleic acid sequences' effects on the aggregation of model protein substrates. In general, these studies are well conducted and controlled.

We are glad that the reviewer thought that the quality of work was good.

However, many of the results presented here are not surprising. Prior work from the Bardwell lab (on which Horowitz is a coauthor) has already shown that that DNA and RNA exhibit potent chaperone activity in vitro.

The reviewer correctly points out that nucleic acids have been shown to have strong chaperone activity in vitro before. The primary point in this paper is that this property is sequence/structure specific, which is a prerequisite to test whether nucleic acids can be chaperones in cells.

Although it is claimed that the sequence-dependence to this activity has been discovered (e.g. G-quads have high activity), whether the mechanisms found in vitro are also active in vivo remains undetermined. Indeed, it remains uncertain whether these sequences would act as chaperones in cells. Without in vivo evidence to support the in vitro data, my enthusiasm for this paper is very low. Moreover, the mechanism of anti-aggregation is proposed to be protein:nucleic acid oligomerization. Given that soluble, misfolded protein oligomers can be highly damaging species in neurodegenerative disease, it could very well be that these species are cytotoxic and that this nucleic-acid chaperone activity is counterproductive in cells. However, this possibility is not studied, which further weakens this paper. Indeed, it is not clear that holding proteins in a misfolded oligomeric state is any better for cells than formation of aggregates when in many cases protein aggregates can be cytoprotective. Overall, the lack of in vivo studies to corroborate the in vitro studies here severely weaken this paper, which would be better suited to a more specialized biochemistry journal.

The reviewer does also correctly point out that there is currently little to no evidence of whether these sorts of activities carry through into cells. As this was the major point brought up by both reviewers, we have addressed this point with new experiments that can be seen in Fig. 4, Fig. S4, and the new section of the results entitled “Chaperone Activity in *E. coli*”. These new experiments test whether the sequences that we most highly verified *in vitro* are able to increase the folded amount of a fluorescent protein that usually struggles to fold in *E. coli*. This work is built off previous work by multiple other labs showing these properties for GFP, and that GFP's interaction with chaperones in *E. coli* improves its folding. With this approach, we show that the nucleic acids that our *in vitro* screen shown to have high activity also help the folding environment for our test protein in cells, unlike ssDNAs that we showed had low activity *in vitro*. Please note that this new section is not intended to be an in depth study of the mechanism in cells, as this would be a considerably longer investigation and beyond the scope of this study, which is primarily on identifying sequence dependence.

Referee #3:

The authors investigate the "holdase" activity (i.e. ability to prevent protein aggregation) of different DNA oligonucleotides (< 20 nt in length) in various protein aggregation assays. They find that oligos with sequences having a propensity to form G-quadruplexes tend to have the highest apparent holdase activity. The observations are interesting, but I have several concerns about the findings and presentation, and the biological significance of the findings seems quite unclear. My specific concerns are as follows.

1) A major problem with the manuscript is that there is no effort made to address the biological significance of the findings. First, the authors provide no discussion, not even a purely theoretical one, of how biologically plausible it is that G-quadruplexes within cells could have the "holdase" activity demonstrated in vitro.

Our original discussion was quite brief, and we have significantly expanded it in this submission, including more context of how quadruplexes could be in relevant locations to have these functions in cells.

Second, and more importantly, for a high impact journal like EMBO Reports, my feeling is that some sort of direct biological evidence for the importance of the activity is essential. I list additional concerns below, but regardless of whether they are addressed or not, I think this principal concern is of central importance.

Like Reviewer 1, Reviewer 3 cites the lack of in cell evidence as being the primary issue with the work. The description below is the same as that provided above in the comments for Reviewer 1:

The reviewer does also correctly point out that there is currently little to no evidence of whether these sorts of activities carry through into cells. As this was the major point brought up by both reviewers, we have addressed this point with new experiments that can be seen in Fig. 4, Fig. S4, and the new section of the results entitled "Chaperone Activity in *E. coli*". These new experiments test whether the sequences that we most highly verified *in vitro* are able to increase the folded amount of a fluorescent protein that usually struggles to fold in *E. coli*. This work is built off previous work by multiple other labs showing these properties for GFP, and that GFP's interaction with chaperones in *E. coli* improves its folding. With this approach, we show that the nucleic acids that our *in vitro* screen shown to have high activity also help the folding environment for our test protein in cells, unlike ssDNAs that we showed had low activity *in vitro*. Please note that this new section is not intended to be an in depth study of the mechanism in cells, as this would be a considerably longer investigation and beyond the scope of this study, which is primarily on identifying sequence dependence.

2) The description of the G-quadruplex-forming oligos needs to be improved. In particular, the authors indicate that these oligos contain the consensus GGGGGNT (and another similar one), but if they were to contain only one of such motif, this would require the presence of at least two such oligos in any G-quadruplex that is formed, and it isn't clear that such G-quadruplexes would form efficiently under the conditions employed (e.g. to form dimeric or tetrameric structures requires sufficiently high oligo concentrations and time). Fortunately, visual inspection of the sequences of the oligos in the supplemental material makes it very clear that

most of them do indeed have a high capacity to form intramolecular G-quadruplexes (i.e. each oligo can form its own G-quad, most of them with stacks of only two guanine quartets, although in several cases dimers or tetramers would need to form), and so, together with the evidence in Fig. 2A and B, I do believe that many/most of the oligos in question do have G-quadruplex forming potential. I suggest that the authors: a) describe the actual oligo sequences in the main text so that readers are not confused about this issue,

We apologize for our lack of clarity in not including a table with the highly used sequences in the main text, which we have improved by adding the new Table 1 that includes this information.

b) describe their annealing conditions for their CD experiments properly so that readers familiar with G-quadruplex biochemistry will be able to understand the findings in 2A,

This is an interesting point. Our original screen did not have an annealing step due to the technical challenges it would have caused. As a result, we did not perform annealing before this CD experiment, so as to be consistent with the amount of quadruplex that would have been present in our initial screen. We have now performed more CD experiments to compare annealed vs non-annealed. For 536, the spectra are nearly identical, and for 576, they are fairly similar. However, for 359, the spectrum changes upon annealing, as well as changes with relatively small changes in buffer. This result suggests that 359's structure is less stable and able to undergo structural rearrangements (now shown in Fig S2). To check whether the annealed version is also chaperone active, we performed a new set of aggregation experiments to compare annealed vs un-annealed sequence 359. These experiments found that the chaperone activity decreased slightly, but that it was still an excellent chaperone after annealing. These result suggest that there may be a dynamic or heterogeneous structural element to the chaperone activity, which would not be surprising, as a large degree of dynamics is a common property of most known chaperone proteins. However, fully investigating this aspect will be challenging and will require extensive further studies that are outside the scope of this manuscript.

and C) use one of the various algorithms that are available (e.g. see Nucleic Acids Res. 2016 Feb 29; 44(4): 1746-1759) so as to more completely characterize and describe the G-quadruplex forming potential of the oligos they have found.

We have now included this information in the new Table 1 for each oligo that is used past the initial screening stage.

3) It's interesting that the 3 oligos that were characterized all appear to form parallel G-quadruplexes, but it's hard to know if this pattern is by chance given the small number of oligos tested. It would be interesting to test systematically the relative holdase activities of oligos that form parallel vs anti-parallel vs mixed (e.g. 3+1 hybrid) G-quadruplex folds, and to test whether G-quadruplexes with different numbers of stacked quartets (at least 2 vs 3) have different activities.

With the new CD experiments, we can now see that for 359, it is not always in a parallel quadruplex conformation, and that for 576, it is likely at least partially a mixture of states as well. We plan in the future to investigate this aspect more fully.

4) *The holdase assays seem perhaps more complex than implied, and I don't feel that enough information is provided to understand how the oligos might be exerting their effects. For example, for the CS aggregation assay used for the screen, what does it mean that one oligo allows 80% aggregation and another allows 20%? - e.g. what is the dose-response relationship for oligo concentration vs holdase activity? It's hard to understand how much better the G-quad forming oligos are than non-G-quad without such context.*

The dose dependence was included before in the supplemental material, but it was not clearly cited in the text, which has been remedied in this version (now Fig. S3).

Moreover, the authors describe holdase activity as "preventing protein aggregation", but doesn't the promotion of protein oligomerization (Fig 4) by the G-quads argue that it isn't that simple? It seems to me that the G-quads actually promote small aggregates to form --- i.e. they promote aggregates that are large enough to cause a small increase light scattering, but prevent further growth into aggregates that are as large as the aggregates formed in their absence (and so they scatter light less and are not spun down easily).

The reviewer is correct that due to the oligomerization, the initial screen is not as simple as it seems on the surface. We delve into this topic much more deeply for nucleic acids in general in our recent paper (Litberg et al. Biophys J. 2020;118(1):162–171. doi:10.1016/j.bpj.2019.11.022), and so we have only summarized similarities that occur in this work. We have added more at this point in the results to help clarify it:

“XXX”

By the way, how large are these small aggregates (the text under the scale bars in Fig 4C is too small to see), and what might this say about what might be happening in vivo?

Since the previous submission, we have collected new TEM data showing that the oligomerization is similar to cases we have observed in the past (Litberg et al. Biophys J. 2020;118(1):162–171. doi:10.1016/j.bpj.2019.11.022), and we have also made the scale bar more clear.

Dear Scott,

Thank you for your patience while your revised manuscript was peer-reviewed at EMBO reports. We have now received the comments from referees 1 and 3, as well as from a new referee (#2).

As you will see, referee 1 remains unconvinced, referee 2 is positive, and referee 3 still has remaining concerns with your revised study. I have discussed all reports with my colleagues here, and we have decided that we can offer to publish your manuscript if you can successfully address all concerns raised by referees 2 and 3. I notice the comment on the specificity of the G4 antibody that does not seem to be a reliable marker for G4 structures. This is an important concern.

If you think that you can address all concerns, please send us a revised manuscript as soon as possible. Please also co-submit a detailed point by point response. Acceptance of the manuscript will depend on positive support by referee 3.

I am looking forward to receiving a final manuscript.

Referee #1:

Although the authors have partially addressed some of my prior concerns, several issues still remain. First, is novelty. It is already known that DNA and RNA exhibit potent chaperone activity in vitro. Second, it has remained unclear whether the mechanism of anti-aggregation involving protein:nucleic acid oligomerization would generate toxic soluble species. This issue continues to go unaddressed in the present work.

Referee #2:

The manuscript by Begeman et al. investigates the role of ssDNA in avoiding protein aggregation.

The idea is original and the paper should in my opinion stimulate the community on the important problem of holdase activity of nucleic acids. I recommend the paper for publication upon some modification.

Comments:

"To determine the sequence specificity of the holdase activity of nucleic acids, we measured light scattering and turbidity via absorbance in a thermal aggregation assay (Fig. 1A) for 312 nucleic acid

sequences (Fig. 1B). "

For the most relevant sequences it would have been useful to have different protein:DNA ratios, rather than only 1:2

"This motif contains five consecutive guanines followed by any base and then thymine. A similar G-rich motif (consensus pattern: BGGSTGAT)"

This motif represent 1/3 of the whole sequence. I wonder if it would still retain the same presumed holdase activity if inserted in a longer sequence...maybe a comment?

"The CD spectra showed distinct peaks at 260 and 210 nm, with a trough at 245 nm, indicative of parallel quadruplex formation (Fig. 2A) (18), and distinct from a control sequence (sequence 42) that had poor chaperone activity and no polyG motif. "

I have two comments here

1. All CD spectra (a part from the control one) show the profile of parallel G-quadruplex. Have the author considered if the orientation of the G-quadruplex may be relevant?
2. I do not fully understand the CD of the control. I would appreciate a sentence of explanation on what it may represent.

"Of note, sequence 359's CD spectrum changed depending on the initial buffer conditions and upon annealing, suggesting that it is able to sample other conformations (Fig S2)."

Tertiary structure in polynucleotides is highly dependent on the ion selected and on the concentration.

Buffer used for aggregation assays and for CD are different, Why? The concentration of oligonucleotide is 1 μ M for the aggregation assay, while the CD (because of the limited sensitivity of the instrument) is recorded at 25 μ M. However, how can we make sure that we can still obtain G-quadruplex at a much lower concentration? It would also be very interesting to know the potential structural variations of nucleic acids when in the presence of the protein substrate. Doing this with CD is not possible (the signal of the protein would cover the one for DNA). Is there any other way to check this?

" In other words, could the activity arise from any DNA with greater structure than ssDNA? To test this possibility, we tested the holdase activity of 24 duplexed sequences to compare directly with their single-stranded counterparts. Overall, the differences were small, and in many cases statistically insignificant (Fig. 2D). These experiments"

An even more interesting control would have been to check anti-parallel or asymmetric G-quadruplex.

"However, the single-stranded sequences demonstrated little to no activity for all of these proteins (Fig. 3). These data strongly suggest that the holdase activity displayed by quadruplex sequences is general, while also unique to quadruplex-forming sequences."

Any idea of the potential nucleotide-binding ability of the other tested proteins? Of what sort of Kd are we talking about? Please comment.

"As controls, we compared both against empty vector, as well as sequence 42, which displayed little to no in vitro activity. Of note, unlike our in vitro testing, in this experiment, the quadruplex-containing sequences are expressed as RNA from a pBAD33 plasmid."

It could have been interesting to test the potential structural polymorphism of the RNA versions of these sequences. A confirmation via CD would be sufficient. It would be also interesting to see the differences in aggregation inhibition between DNA and RNA versions of the same sequences.

"Finally, negative stain TEM imaging showed that the quadruplexes caused the formation of protein oligomers (Fig. 5C). The morphology of these oligomers was similar to previously observed oligomers formed in the presence of bulk DNA, suggesting similar mechanisms to those observed previously (32)."

In the turbidity assay there is no oligomer formation, while this can be seen under TEM. I am still a bit worried about an effect of potential precipitation, which would be ideal to exclude.

Finally i do not understand this in the reply to Referee 2:

"The reviewer is correct that due to the oligomerization, the initial screen is not as simple as it seems on the surface. We delve into this topic much more deeply for nucleic acids in general in our recent paper (Litberg et al. Biophys J. 2020;118(1):162-171. doi:10.1016/j.bpj.2019.11.022), and so we have only summarized similarities that occur in this work. We have added more at this point in the results to help clarify it: XXX".

The main point here is: are these oligomers evolving towards big aggregates or stay in their state for long?

Gian Gaetano Tartaglia

Referee #3:

The manuscript has been improved, and overall I find this to be a very interesting and potentially important story, but there are still problems:

1. The new data in Figure 4 showing that the G4-forming oligos can enhance RFP fluorescence in *E. coli* is very encouraging. However, a critical control is missing. Guo and Bartel (2016, Science 353) showed that the expression of G4-forming RNA is apparently toxic in *E. coli*, as revealed by a slowed growth rate. This was particularly pronounced if the G4 was translated. There is not enough information provided in the current manuscript to know whether the G4 RNAs tested were translated or not (and such details should be provided), but regardless, it is quite possible that the reason G4-forming oligos are enhanced RFP fluorescence is because they induce stress responses that upregulate native chaperones. In other words the effects of the G4 RNAs are indirectly mediated by *E. coli* chaperones. An ideal way to address this is to test the dependence on such chaperones, but I don't know how feasible this is (i.e. can you delete enough to really test the idea?), but at a bare minimum, it is critical for the authors to carefully measure the growth rate of the G4 RNA-expressing vs non-expressing strains. If there is no growth inhibition by G4 RNA under

the conditions of their experiments, then I would be satisfied.

2. There is still a lack of careful attention to important details.

For example, Table 1 lists 14 G4-forming oligo sequences based on G4 Hunter scores. These are said to correspond to oligo sequences in Figure 3, but the figure shows 15 sequences, and the Fig. 3 legend says 16 sequences are boxed but only 15 are. And Table 1 lists 10 non-forming G4 sequences said to correspond to Figure 3, but the figure shows 9. To add to confusion, in the table these are given letters, but they are numbered in Figure 3. My guess is that Seq353 (aka oligo O, aka #15) and Seq63 (aka oligo P, aka #16) can form G-quadruplexes to some degree (despite the G4 Hunter prediction), and so the data actually support the authors' hypothesis. It would be helpful if the authors could clarify their presentation and thoughts about this. It would also be helpful to test whether oligos O and P can form G4s (e.g. via CD) - I bet they do, and G4 Hunter isn't a perfect predictor of G4 forming potential.

Another example: I asked for details of how annealing was performed. They now say the oligos were "heated to 94 degrees for 2 minutes and allowed to cool to room temperature with the internal fan of the heating block". The obvious question is: what was the time course of the cooling? Apparently it was too fast, because their new data on "annealed" oligos shows that they have less G4 conformation than non-annealed. This is backwards from the point of annealing. The other question is how did they actually prepare their non-annealed? Did they resuspend them in buffer (at what temperature) and use them immediately or after some period of time? It would be helpful for the authors to read some basic information about how this sort of work is done.

Similarly, the authors indicate that they have added new information in the Results to clarify my concern about the complexity of the holdase assay, but I don't see this. Their response letter indicates that the new text consists of "XXX" - literally. Perhaps they forgot to address this issue?

3) The authors have ignored my previous concern #3

(which was:

"3) It's interesting that the 3 oligos that were characterized all appear to form parallel G-quadruplexes, but it's hard to know if this pattern is by chance given the small number of oligos tested. It would be interesting to test systematically the relative holdase activities of oligos that form parallel vs anti-parallel vs mixed (e.g. 3+1 hybrid) G-quadruplex folds, and to test whether G-quadruplexes with different numbers of stacked quartets (at least 2 vs 3) have different activities.")

This could be argued to be outside the scope of the manuscript (I guess), but it would have been nice for the authors to address my point.

4) I asked previously for the authors to discuss the biological plausibility of their model. I'm not saying it's not plausible (in fact I'm an enthusiast when it comes to G-quadruplex biology). But, e.g., it is essential for the authors to acknowledge the very important findings in the Guo and Bartel paper mentioned above (i.e. if G4s are so important as chaperones in vivo, why can't they be observed)? [Incidentally, if these authors want to invoke evidence based on the widely used BG4 antibody, they should ask themselves whether or not it is a reliable indicator of G4 presence (as the vast majority of biologists seem to believe). E.g. they should read: Ray et al., ACS Chem. Biol. 2020,

15:925-935 (fig 3 in particular) which demonstrates the antibody recognizes most ssDNA sequences as readily as those that can form G4s, and also recognize that anti-G4 antibodies can induce G4 folds artifactually].

Referee #1:

Although the authors have partially addressed some of my prior concerns, several issues still remain. First, is novelty. It is already known that DNA and RNA exhibit potent chaperone activity *in vitro*. Second, it has remained unclear whether the mechanism of anti-aggregation involving protein:nucleic acid oligomerization would generate toxic soluble species. This issue continues to go unaddressed in the present work.

The reviewer seemingly did not see that for this revision we added an entire new section of the paper with work in *E. coli*, the first such experiments *in vivo*, in part to address this novelty issue.

Referee #2:

The manuscript by Begeman et al. investigates the role of ssDNA in avoiding protein aggregation.

The idea is original and the paper should in my opinion stimulate the community on the important problem of holdase activity of nucleic acids. I recommend the paper for publication upon some modification.

Comments:

"To determine the sequence specificity of the holdase activity of nucleic acids, we measured light scattering and turbidity via absorbance in a thermal aggregation assay (Fig. 1A) for 312 nucleic acid sequences (Fig. 1B). "

For the most relevant sequences it would have been useful to have different protein:DNA ratios, rather than only 1:2

In our previous submission, we had tested multiple concentrations for our best sequence, 359, showing a dependence on concentration (now Fig EV2B). In this version, we now include experiments for all three of our most-tested quadruplex sequences to test aggregation prevention as a function of concentration, again including sequence 42 as a negative control (Fig EV2A). The new data showed that all three quadruplex-forming sequences had strong concentration-dependent activity, unlike sequence 42, and is pasted below:

Fig EV2. Concentration dependence of chaperone nucleic acid activity.

A Percent aggregation in thermal denaturation assay with varying concentrations of select quadruplex forming sequences (seq 359, 536 and 576) and negative control sequence (seq 42). Concentrations are ssDNA strand to protein ratios of: 0.5:1, 1:1, 2:1, 4:1, and 8:1.

B Chemical aggregation test of the concentration dependence of holdase activity of sequence 359, the best-performing holdase sequence. Concentration ratios are citrate synthase:DNA strand.

"This motif contains five consecutive guanines followed by any base and then thymine. A similar G-rich motif (consensus pattern: BGGSTGAT)"

This motif represent 1/3 of the whole sequence. I wonder if it would still retain the same presumed holdase activity if inserted in a longer sequence...maybe a comment?

In this study, we did not consider length as a variable, but it is an important one to consider in future work. We have added a note to this effect in the discussion,

"Despite the number of sequences tested here, many quadruplex variants remain for further future exploration. Similarly, the tests here were performed with a single length of nucleic acid (20 bases), whereas greater topology variance could likely be achieved with longer sequences. Future work could greatly refine the proof-of-principle concepts investigated here."

"The CD spectra showed distinct peaks at 260 and 210 nm, with a trough at 245 nm, indicative of parallel quadruplex formation (Fig. 2A) (18), and distinct from a control sequence (sequence 42) that had poor chaperone activity and no polyG motif. "

I have two comments here

1. All CD spectra (a part from the control one) show the profile of parallel G-quadruplex. Have the author considered if the orientation of the G-quadruplex may be relevant?

We have now included a new experiment to test this possibility. We chose multiple quadruplexes of known topology based on the literature, and tested their ability to prevent aggregation. In short, the orientation was important, with anti-parallel quadruplexes having especially poor chaperone activity, while 3+1 mixed quadruplexes had good chaperone activity. This data is now shown in Fig 2D:

D

D Comparing holdase activity of different quadruplex-containing sequence of known topology (21–29).

2. I do not fully understand the CD of the control. I would appreciate a sentence of explanation on what it may represent.

The negative control sequence, 42, is a ssDNA that has negligible probability of forming quadruplexes, or even self-complementarity, and it displayed negligible chaperone activity in our aggregation assays. Its CD spectrum is reminiscent of other A-rich ssDNA sequences ([https://doi.org/10.1016/S0022-2836\(66\)80122-5](https://doi.org/10.1016/S0022-2836(66)80122-5), <https://doi.org/10.1002/chem.201704338>, <https://doi.org/10.3390/sym11040567>). They are not that different from helical spectra, due to A-rich single-stranded sequences having a high degree of helical character ([https://doi.org/10.1016/S0022-2836\(66\)80122-5](https://doi.org/10.1016/S0022-2836(66)80122-5), [https://doi.org/10.1016/S0022-2836\(66\)80121-3](https://doi.org/10.1016/S0022-2836(66)80121-3), <https://doi.org/10.1093/nar/gkr833>).

"Of note, sequence 359's CD spectrum changed depending on the initial buffer conditions and upon annealing, suggesting that it is able to sample other conformations (Fig S2)."

Tertiary structure in polynucleotides is highly dependent on the ion selected and on the concentration.

Buffer used for aggregation assays and for CD are different, Why?

The initial aggregation assay conditions were based off of standard aggregation assay conditions for citrate synthase from previous chaperone studies that also used this assay. Due to the touchy nature of aggregation experiments and the need for maximum reproducibility, we did not want to deviate from these conditions for the screen. However, these conditions have substantial amount of HEPES, which is incompatible with CD spectroscopy, necessitating a switch in buffer. For this version, we have added another buffer condition that is more similar to that of the screen (potassium phosphate as opposed to sodium phosphate) to favor quadruplex formation, and mimic the potassium concentrations used with the HEPES buffer in the initial screen. The results of these CD experiments with potassium phosphate can be found in Fig EV3:

Fig EV3. CD spectra of select 20mers (DNA (A) and (B), and RNA (C) and (D)) in 10 mM pH 7.5 potassium phosphate.

The use of potassium phosphate here was to better mimic the initial aggregation screen in which potassium was used in the HEPES buffer.

A, C The thermal stability of quadruplex-containing sequences as measured by CD spectroscopy where each line represents a wavelength scan at the indicated temperature.

B, D The secondary structure of the same quadruplex-containing sequences at 25° C prior to annealing or after annealing at 25° C.

The concentration of oligonucleotide is 1 μ M for the aggregation assay, while the CD (because of the limited sensitivity of the instrument) is recorded at 25 μ M. However, how can we make sure that we can still obtain G-quadruplex at a much lower concentration?

The reviewer is correct that the CD experiments require much higher concentrations than used in our aggregation assays. For this reason, we also employed the parallel quadruplex binding fluorophore NMM-IX (Fig 2C) to check whether quadruplex formation still occurred at the concentration range used in aggregation assays. The change in signal indicated that there was a

substantially greater degree of quadruplex formation in the predicted quadruplexes than in the single-stranded control. The figure panel is pasted below for convenience:

C

C NMM fluorescence measured at 610 nm.

It would also be very interesting to know the potential structural variations of nucleic acids when in the presence of the protein substrate. Doing this with CD is not possible (the signal of the protein would cover the one for DNA). Is there any other way to check this?

As to whether the nucleic acid changes conformation when bound to the protein, this is very difficult for chaperones in general, especially for those that form oligomers, and we believe it beyond the scope of the manuscript and would be its own separate study.

" In other words, could the activity arise from any DNA with greater structure than ssDNA? To test this possibility, we tested the holdase activity of 24 duplexed sequences to compare directly with their single-stranded counterparts. Overall, the differences were small, and in many cases statistically insignificant (Fig. 2D). These experiments"

An even more interesting control would have been to check anti-parallel or asymmetric G-quadruplex.

As mentioned above, we have now added this experiment, which does show differences between different quadruplex topologies in Fig 2D:

D

D Comparing holdase activity of different quadruplex-containing sequence of known topology (21–29).

"However, the single-stranded sequences demonstrated little to no activity for all of these proteins (Fig. 3). These data strongly suggest that the holdase activity displayed by quadruplex sequences is general, while also unique to quadruplex-forming sequences."

Any idea of the potential nucleotide-binding ability of the other tested proteins? Of what sort of K_d are we talking about? Please comment.

Of the proteins tested for generality here, the only one with previously identified nucleic acid binding activity is LDH, which tightly binds multiple forms of nucleic acids. For the other proteins, no previous nucleic acid binding activity has been reported to the best of our knowledge. A rigorous K_d test is not possible in binding modes investigated here, because the protein aggregates, and because the CD spectra of the bound protein (Fig S3) suggests that the protein is bound in a non-native conformation. This scenario is further complicated by the bound state forming co-oligomers. Taken together, it is very difficult to make meaningful estimates of what the K_d could be in these cases.

"As controls, we compared both against empty vector, as well as sequence 42, which displayed little to no in vitro activity. Of note, unlike our in vitro testing, in this experiment, the quadruplex-containing sequences are expressed as RNA from a pBAD33 plasmid."

It could have been interesting to test the potential structural polymorphism of the RNA versions of these sequences. A confirmation via CD would be sufficient. It would be also interesting to see the differences in aggregation inhibition between DNA and RNA versions of the same sequences.

We have now tested the RNA cognates of these sequences, which also generally form parallel quadruplexes. The spectra can now be found in Fig EV3C & D.

Fig EV3. CD spectra of select 20mers (DNA (A) and (B), and RNA (C) and (D)) in 10 mM pH 7.5 potassium phosphate.

The use of potassium phosphate here was to better mimic the initial aggregation screen in which potassium was used in the HEPES buffer.

A, C The thermal stability of quadruplex-containing sequences as measured by CD spectroscopy where each line represents a wavelength scan at the indicated temperature.

B, D The secondary structure of the same quadruplex-containing sequences at 25° C prior to annealing or after annealing at 25° C.

Because 359's CD spectra showed the largest change in quadruplex topology and stability upon switching from DNA to RNA, we tested the aggregation prevention capability of its RNA cognate in vitro, which also showed strong aggregation prevention capabilities, as shown in Fig EV4:

Fig EV4. Comparison of holdase activity of RNA and DNA counterparts of Sequence 359 using citrate synthase heat denaturation.

Data presented as means \pm standard error (n = 3).

"Finally, negative stain TEM imaging showed that the quadruplexes caused the formation of protein oligomers (Fig. 5C). The morphology of these oligomers was similar to previously observed oligomers formed in the presence of bulk DNA, suggesting similar mechanisms to those observed previously (32)."

In the turbidity assay there is no oligomer formation, while this can be seen under TEM. I am still a bit worried about an effect of potential precipitation, which would be ideal to exclude.

In the heat-denaturation turbidity experiment, we do not see clear oligomer formation, but this is probably due to the small size of the oligomers. In the chemical denaturation experiments, we do see an initial jump in light scattering (Fig 2A), strongly suggestive of oligomer formation, which prompted the TEM experiments. However, in the equivalent spin-down assay (Fig S4, pasted below), the best of the sequences (seq 359) completely prevent any detectable pellet, suggesting that the oligomers are too small to precipitate under these conditions.

Fig S4. Prevention of protein aggregation in chemical spin down assay using citrate synthase.

Sequences 359, 536, and 576 all displayed holdase activity and contain a polyG motif. Sequence 42 was used as a negative control, as it performed poorly as a holdase chaperone and did not contain a polyG motif.

We have made changes to the results section to help illustrate these points more clearly:

“Further spin down assays in chemically denaturing conditions suggest that these oligomers tend to not form large aggregates (Appendix Fig S4)”

Finally, I do not understand this in the reply to Referee 2:

"The reviewer is correct that due to the oligomerization, the initial screen is not as simple as it seems on the surface. We delve into this topic much more deeply for nucleic acids in general in our recent paper (Litberg et al. Biophys J. 2020;118(1):162-171. doi:10.1016/j.bpj.2019.11.022), and so we have only summarized similarities that occur in this work. We have added more at this point in the results to help clarify it: XXX".

We had intended to paste a section of the paper in here that we added to address this point, and have done so below:

“These data are highly reminiscent of the pattern we observed recently in which nucleic acids could prevent protein aggregation by promoting protein:nucleic acid oligomerization (32). In this previous study, we found that bulk nucleic acids at high concentration could prevent protein aggregation under extreme conditions through the formation of protein:nucleic acid oligomers that could be controlled by varying nucleic acid concentration (32). While the best quadruplex-containing sequences were considerably more efficient than the bulk DNA used previously, the initial jump in light scattering hinted that the initial holdase assays could have provided an overly simplistic interpretation of their mechanism, and that oligomerization could be at least partially responsible for their activity.”

The main point here is: are these oligomers evolving towards big aggregates or stay in their state for long?

As discussed above, we do not see evidence of continuing evolution after oligomer formation into larger species, as seen in Appendix Fig S4. In the case of bulk DNA that we characterized previously (albeit at much higher nucleic acid concentrations), we found that oligomers were stable at room temperature for weeks without apparent change.

Gian Gaetano Tartaglia

Referee #3:

The manuscript has been improved, and overall I find this to be a very interesting and potentially important story, but there are still problems:

1. The new data in Figure 4 showing that the G4-forming oligos can enhance RFP fluorescence in *E. coli* is very encouraging. However, a critical control is missing. Guo and Bartel (2016, *Science* 353) showed that the expression of G4-forming RNA is apparently toxic in *E. coli*, as revealed by a slowed growth rate. This was particularly pronounced if the G4 was translated. There is not enough information provided in the current manuscript to know whether the G4 RNAs tested were translated or not (and such details should be provided), but regardless, it is quite possible that the reason G4-forming oligos are enhanced RFP fluorescence is because they induce stress responses that upregulate native chaperones. In other words the effects of the G4 RNAs are indirectly mediated by *E. coli* chaperones. An ideal way to address this is to test the dependence on such chaperones, but I don't know how feasible this is (i.e. can you delete enough to really test the idea?), but at a bare minimum, it is critical for the authors to carefully measure the growth rate of the G4 RNA-expressing vs non-expressing strains. If there is no growth inhibition by G4 RNA under the conditions of their experiments, then I would be satisfied.

To confirm the reviewer's suspicions, yes, it is not possible to delete enough chaperones to prevent all indirect effects.

On the question of translation of the G4 RNA and resulting toxicity, the pBAD33 vector used to express the G4 RNA lacks a ribosome binding site, which should prevent significant levels of translation. We would therefore not expect growth inhibition. We then tested this hypothesis as suggested by the reviewer. As shown in Fig S2 and pasted below, the introduction of the G4 RNA had no discernable effect on the growth rate of the *E. coli*.

Fig S2. Growth curves of *E. coli* MC4100(DE3) in the presence or absence of G-quadruplex-containing sequences.

Absorbance at 600nm of cultured *E. coli* cells was measured for 19 hours, with the induction of GroEL, Seq42, Seq359, and Seq576. Non-induced (NI) and induced (IN) Empty vector and non-induced GroEL were used as negative controls. The experiment was performed in triplicate; error bars are standard deviation.

2. There is a still a lack of careful attention to important details.

For example, Table 1 lists 14 G4-forming oligo sequences based on G4 Hunter scores. These are said to correspond to oligos sequences in Figure 3, but the figure shows 15 sequences, and the Fig. 3 legend says 16 sequences are boxed but only 15 are. And Table 1 lists 10 non-forming G4 sequences said to correspond to Figure 3, but the figure shows 9. To add to confusion, in the table these are given letters, but they are numbered in Figure 3. My guess is that Seq353 (aka oligo O, aka#15) and Seq63 (aka oligo P, aka #16) can form G-quadruplexes to some degree (despite the G4 Hunter prediction), and so the data actually support the authors' hypothesis. It would be helpful if the authors could clarify their presentation and thoughts about this.

We apologize- there was a mistake in the figure and table previously that led to this confusion, but they should now be consistent. We have also taken the reviewer's suggestion and changed the numbering to lettering in the figure to make it consistent with the table, as seen below:

It would also be helpful to test whether oligos O and P can form G4s (e.g. via CD) - I bet they do, and G4 Hunter isn't a perfect predictor of G4 forming potential.

We have added new notes in both the figure legend and main text that our quadruplex-determination for this experiment was using G4Hunter, and so it may not be perfect in each case. We additionally added this line to the results section:

“Of note, two sequences (O and P) that G4Hunter did not predict as having high quadruplex probability, but had significant holdase activity, do have substantial guanine content and could potentially still form quadruplexes despite being listed as ssDNA here.”

Another example: I asked for details of how annealing was performed. They now say the oligos were "heated to 94 degrees for 2 minutes and allowed to cool to room temperature with the internal fan of the heating block". The obvious question is: what was the time course of the cooling? Apparently it was too fast, because their new data on "annealed" oligos shows that they have less G4 conformation than non-annealed. This is backwards from the point of annealing. The other question is how did they actually prepare there non-annealed? Did they resuspend them in buffer (at what temperature) and use them immediately or after some period of time? It would be helpful for the authors to read some basic information about how this sort of work is done.

The reviewer pointed out that the annealing did not work as expected in that it appeared that the quadruplex formation did not increase, and in some cases it decreased; this was due to our buffer conditions. In our CD experiments, we were attempting to maintain the same CD conditions we had used in a previous publication for comparability where we saw oligomerization, but these conditions were not very conducive to quadruplex formation, nor were they very similar to the

aggregation assay conditions used here. In this revision, we have performed the CD experiments again, this time using potassium phosphate in place of sodium phosphate, which is a closer analogue to what was used in the aggregation assays due to the presence of potassium, and would be expected to increase quadruplex formation upon annealing. Indeed, this was the case with the new experiments, as can be seen in Fig EV3, pasted below:

Fig EV3. CD spectra of select 20mers (DNA (A) and (B), and RNA (C) and (D)) in 10 mM pH 7.5 potassium phosphate.

The use of potassium phosphate here was to better mimic the initial aggregation screen in which potassium was used in the HEPES buffer.

A, C The thermal stability of quadruplex-containing sequences as measured by CD spectroscopy where each line represents a wavelength scan at the indicated temperature.

B, D The secondary structure of the same quadruplex-containing sequences at 25° C prior to annealing or after annealing at 25° C.

As to the question of the non-annealed samples, these were prepared by simple re-suspension of the initial DNA in buffer. We have now measured the annealing time as well and placed this information in the methods. The annealing process took 30 minutes to cool.

Similarly, the authors indicate that they have added new information in the Results to clarify my concern about the complexity of the holdase assay, but I don't see this. Their response letter indicates that the new text consists of "XXX" - literally. Perhaps they forgot to address this issue?

We apologize for this oversight- we had put in new wording to address this point, but forgot to paste it in the response to reviewers. The new wording is below:

“These data are highly reminiscent of the pattern we observed recently in which nucleic acids could prevent protein aggregation by promoting protein:nucleic acid oligomerization (32). In this previous study, we found that bulk nucleic acids at high concentration could prevent protein aggregation under extreme conditions through the formation of protein:nucleic acid oligomers that could be controlled by varying nucleic acid concentration (32). While the best quadruplex-containing sequences were considerably more efficient than the bulk DNA used previously, the initial jump in light scattering hinted that the initial holdase assays could have provided an overly simplistic interpretation of their mechanism, and that oligomerization could be at least partially responsible for their activity.”

3) The authors have ignored my previous concern #3

(which was:

"3) It's interesting that the 3 oligos that were characterized all appear to form parallel G-quadruplexes, but it's hard to know if this pattern is by chance given the small number of oligos tested. It would be interesting to test systematically the relative holdase activities of oligos that form parallel vs anti-parallel vs mixed (e.g. 3+1 hybrid) G-quadruplex folds, and to test whether G-quadruplexes with different numbers of stacked quartets (at least 2 vs 3) have different activities.")

This could be argued to be outside the scope of the manuscript (I guess), but it would have been nice for the authors to address my point.

We apologize for not addressing this point directly, but we were able to now to test a large part of this question in the new manuscript. The new Fig 2D (pasted below) shows the aggregation prevention capability of quadruplexes of known topology, and there are strong differences between anti-parallel, parallel, and 3+1 quadruplexes. We believe a detailed follow-up on the many possible detailed changes that could cause these changes to be beyond the scope of this study, but that there are interesting differences between their holdase properties.

D

D Comparing holdase activity of different quadruplex-containing sequence of known topology (21–29).

4) I asked previously for the authors to discuss the biological plausibility of their model. I'm not saying it's not plausible (in fact I'm an enthusiast when it comes to G-quadruplex biology). But, e.g., it is essential for the authors to acknowledge the very important findings in the Guo and Bartel paper mentioned above (i.e. if G4s are so important as chaperones in vivo, why can't they be observed)? [Incidentally, if these authors want to invoke evidence based on the widely used BG4 antibody, they should ask themselves whether or not it is a reliable indicator of G4 presence (as the vast majority of biologists seem to believe). E.g. they should read: Ray et al., ACS Chem. Biol. 2020, 15:925-935 (fig 3 in particular) which demonstrates the antibody recognizes most ssDNA sequences as readily as those that can form G4s, and also recognize that anti-G4 antibodies can induce G4 folds artifactually].

The Guo *et al.* paper was important for contextualizing the many new studies on quadruplexes, and we have explicitly cited in the discussion when talking about more biological significance,

“Similarly, our work here was in *E. coli*, in which quadruplexes form readily, but form less readily in eukaryotes (49).”

We have also added a reference to Ray *et al.*, as suggested by the reviewer, as it does call into question the accuracy of some papers that heavily use the BG4 antibody. Although we do not use the BG4 antibody here, several papers cited in our paper do, and it important to alert the reader to potential inaccuracies in these previous reports:

“However, it is worth bearing in mind that quadruplexes are often localized utilizing the BG4 antibody, which has recently had its specificity called into question (48).”

Dear Scott,

Thank you for the submission of your revised manuscript. I am happy to tell you that referee 3 is satisfied with your last responses. We can therefore in principle accept your study for publication here.

However, a few more minor changes will be required:

- The Appendix needs a table of content with page numbers and the nomenclature needs to be corrected to Appendix Figure S1, etc.
- Appendix Fig S1 needs to specify "n" and the error bars in the figure legend.
- The reference format should only list up to 10 authors followed by "et al" if there are more authors, please correct.
- Fig 1 source data is called Fig 1B and 3 (the title of the excel file). Please upload the source data as one file for each individual main figure. If there is source data for figure 3 in the file, please remove that and upload another file called Fig 3 source data.
- The README source data text file should be included with the relevant source data themselves. Either paste the relevant text into the excel file with the source data, or upload 1 source data file with several pages.

I would like to suggest a few changes to the title and abstract. Please let me know whether you agree with the following:

G-Quadruplexes Act as Sequence-Dependent Protein Chaperones

Maintaining proteome health is important for cell survival. Nucleic acids possess the ability to prevent protein aggregation more efficiently than traditional chaperone proteins. In this study, we explore the sequence specificity of the chaperone activity of nucleic acids. Evaluating over 500 nucleic acid sequences' effects on protein aggregation, we show that the holdase chaperone effect of nucleic acids is sequence-dependent. G-Quadruplexes prevent protein aggregation via quadruplex:protein oligomerization. They also increase the folded protein level of a biosensor in *E. coli*. These observations contextualize recent reports of quadruplexes playing important roles in aggregation-related diseases, such as Fragile X and Amyotrophic lateral sclerosis (ALS), and provide evidence that nucleic acids have the ability to modulate the folding environment of *E. coli*.

I attach to this email a related ms file with comments by our data editors. Please address all comments in the final manuscript.

EMBO press papers are accompanied online by A) a short (1-2 sentences) summary of the findings and their significance, B) 2-3 bullet points highlighting key results and C) a synopsis image that is 550x200-600 pixels large (the height is variable). You can either show a model or key data in the

synopsis image. Please note that text needs to be readable at the final size. Please send us this information along with the revised manuscript.

(The one sentence summary you submitted could be used:

Examination of nucleic acids for sequence dependence of their chaperone activity reveals that G-quadruplexes potentially prevent protein aggregation.)

We can now offer to publish your paper back to back with Denise Sheer's paper. In this case, I would suggest that you both cite each others' work, to make both studies more visible. I will ask Denise to do the same.

I look forward to receiving the final version of your manuscript as soon as possible.

Referee #3:

I'm satisfied with the author's responses to my concerns. I think the findings, particularly the E. coli experiments, are sufficiently novel for publication and that will be of great interest to many readers.

The authors have addressed all minor editorial concerns.

Prof. Scott Horowitz
University of Denver
2155 E Wesley Ave
Room 561
CO 80113
United States

Dear Scott,

I am very pleased to accept your manuscript for publication in the next available issue of EMBO reports. Thank you for your contribution to our journal.

At the end of this email I include important information about how to proceed. Please ensure that you take the time to read the information and complete and return the necessary forms to allow us to publish your manuscript as quickly as possible.

As part of the EMBO publication's Transparent Editorial Process, EMBO reports publishes online a Review Process File to accompany accepted manuscripts. As you are aware, this File will be published in conjunction with your paper and will include the referee reports, your point-by-point response and all pertinent correspondence relating to the manuscript.

If you do NOT want this File to be published, please inform the editorial office within 2 days, if you have not done so already, otherwise the File will be published by default [contact: emboreports@embo.org]. If you do opt out, the Review Process File link will point to the following statement: "No Review Process File is available with this article, as the authors have chosen not to make the review process public in this case."

Should you be planning a Press Release on your article, please get in contact with emboreports@wiley.com as early as possible, in order to coordinate publication and release dates.

Thank you again for your contribution to EMBO reports and congratulations on a successful publication. Please consider us again in the future for your most exciting work.

THINGS TO DO NOW:

You will receive proofs by e-mail approximately 2-3 weeks after all relevant files have been sent to

our Production Office; you should return your corrections within 2 days of receiving the proofs.

Please inform us if there is likely to be any difficulty in reaching you at the above address at that time. Failure to meet our deadlines may result in a delay of publication, or publication without your corrections.

All further communications concerning your paper should quote reference number EMBOR-2019-49735V4 and be addressed to emboreports@wiley.com.

Should you be planning a Press Release on your article, please get in contact with emboreports@wiley.com as early as possible, in order to coordinate publication and release dates.

Corresponding Author Name: Scott Horowitz

Manuscript Number: EMBOR-2019-49735V3